# Identification of the major rabbit and guinea pig semen coagulum proteins and description of the diversity of the REST gene locus in the mammalian clade Glires

Åke Lundwall[1]*, Margareta Persson[1], Karin Hansson[2,3], Magnus Jonsson[4]

**1** Department of Laboratory Medicine, Division of Translational Cancer Research, Lund University, Lund, Sweden, **2** Department of Laboratory Medicine, Division of Clinical Chemistry and Pharmacology, Lund University, Lund, Sweden, **3** Department of Immunotechnology, Lund University, Lund, Sweden, **4** Department of Translational Medicine, Section for Clinical Chemistry, Lund University, Malmö, Sweden

* ake.lundwall@med.lu.se

**Data Availability Statement:** The novel cDNA sequences of Guinea pig Svp5 and rabbit SVP200

## Abstract

The seminal vesicle secretions of guinea pig and rabbit were analyzed for semen coagulum proteins. Using SDS-PAGE we discovered a previously not fully recognized semen coagulum protein, Svp5, in the guinea pig and a single predominant component, SVP200, in the rabbit. Potential genes of these proteins were identified in genome databases by their homology with human and murine genes. The structure of their fullength transcripts was determined using seminal vesicle cDNA and sequencing primers based on genomic sequences. Homology searching indicated that both Svp5 and SVP200 were synthesized from composite genes that were the result of merger between two genes showing homology with human *SEMG2* and *PI3*. For a deeper understanding of the evolution of the genes, we retrieved and analyzed genome sequences from the REST gene loci, encompassing genes of semen coagulum proteins and related rapidly evolving seminal vesicle-transcribed genes, of 14 rodents and 2 lagomorphs. The analysis showed that rodents of the suborders myomorpha, hystricomorpha, and castorimorpha had unique sets of REST genes, whereas sciuromorpha seemed to be lacking such genes. It also indicated a closer relationship between myomorpha and castorimorpha than to rodents of the two other analyzed suborders. In the lagomorph species, the pika appeared to be devoid of REST genes, whereas the rabbit had a single expressed REST gene, SVP200, and two pseudogenes. The structural similarity of semen coagulum proteins in rabbit and hystricomph species suggests that they are closely related. This was also supported by other similarities at their REST gene loci, *e.g.* the finding of a PI3-like gene in the rabbit that also had features in common with *caltrin2* of hystricomorph rodents. The homologies indicate that hystricomorpha may have separated from myomorpha and castorimorpha before the separation of hystricomorpha from lagomorpha.

are deposited with GenBank under accession numbers MT416574 and MT416575.

**Funding:** The study was funded by a grant from Alfred Österlunds stiftelse, Malmö, Sweden http://www.alfredosterlundsstiftelse.se/ The funder had no role in study design, data collection and analysis, decision to publish, or preparation of the manuscript.

**Competing interests:** The authors have declared that no competing interests exist.

## Introduction

The semen of male mammals is formed at ejaculation by the mixing of spermatozoa-rich epididymal fluid with secretions from accessory sex glands. In many species, the newly ejaculated semen appears as a coagulum of varying consistency, which may subsequently liquefy within minutes, as in humans [1], or become stabilized and develop into a post copulatory vaginal plug following mating, as in some rodents [2]. The semen coagulum proteins (SCPs) are secreted by the seminal vesicles at very high concentration and can in many cases be visualized as discrete bands on gels stained for proteins following SDS-PAGE of seminal vesicle secretion (SVS) [3, 4]. Studies on the molecular properties of these proteins, and components related to them, have shown that there are two homologous coagulum proteins, denoted semenogelin 1 (SEMG1) and semenogelin 2 (SEMG2), in semen of humans and many primates [1, 5]. In the SVS of rat and mouse, there are 6 major proteins, denoted Svs1-Svs6, of which Svs1-Svs3 are reported to be coagulum proteins, with Svs2 as the predominating component [4, 6, 7]. The guinea pig seminal vesicles are reported to yield 4 protein components from two major transcripts, one of which generates Svp2 [8], while the other transcript generates a polyprotein precursor of 43 kDa, which is processed to three secreted components; the major coagulum protein Svp1 and the overlapping and almost identical Svp3 and Svp4 [9].

Comparative studies on the structure of primate semenogelins show that they have undergone an unusual and phylogenetically late evolution. Large parts of the molecules consist of poorly defined repeats of 60 amino acid residues (AA) [10]. Differences in the number of repeats have created species specific molecular size variation, *e.g.* the larger number of repeats in chimpanzee SEMG1 makes it almost twice as big as its human counterpart [11]. There are also examples of intra species-size variation due to differences in the number of repeats, *e.g.* an estimated 3% of the human gene pool is lacking one of the repeats present in the predominating form of *SEMG1* [11–13]. There are also additional size heterogeneity introduced by the frequent occurrence of both fixed and polymorphic stop codons affecting both SEMG1 and SEMG2 [11, 14]. Species specific size variation has also been reported for SCPs in closely related rodents, *e.g.* murine Svs1 and Svs2 display differences in the number of 18 and 13–29 AA repeats in rat and mouse, which generates larger coagulum proteins in the rat than in the mouse [15–17].

Studies on the genes of primate and rodent SCPs have demonstrated a common architecture [18]. Most genes of SCPs consist of three exons: The first is a signal peptide-coding exon (SPCE), containing a short stretch of nucleotides (NT) that are non-translated, followed by the sequence encoding the signal peptide and a few AA of the secreted SCP. The second is the major coding exon (MCE), which hold the reminder of the coding NT and a few 3' non-translated NT. The third is the 3' non-translated exon (3NTE), with non-coding NT only, including the poly-adenylation signal. Sequence comparison of the predominant semen coagulum protein in human and mouse, SEMG1 and Svs2, shows an almost complete lack of conserved primary structure. However, conserved nucleotide sequences in the SPCE and 3NTE, as well as in intron and gene flanking sequences, show that *SEMG1* and *Svs2* are homologous in spite of the limited sequence similarity in the MCE and hence also of the protein products.

A similar organization and conservation of NT is also apparent in most other genes of SCPs that have been studied this far. The exception is mouse and rat *Svs1*, which is related to *Aoc1*, the gene encoding diamine oxidase and which has a distinctly different organization [17]. The genes of the small rat and mouse seminal vesicle-secreted proteins Svs4-Svs6 are located at the same chromosome locus as *Svs2*, as are two almost identical Svs3 genes, *Svs3a* and *Svs3b*. These 6 genes also display a similar exon organization and pattern of conservation and in order to emphasize the homology of these genes, in spite of the lacking similarity in primary

structure of their protein products, they were assembled in a gene family denoted the REST gene family, where the acronym stands for Rapidly Evolving Seminal vesicle Transcribed [16, 18]. Two mechanisms, internal exon expansion of the MCE by duplication of small DNA segments in combination with replication slippage, and de novo selection of splice acceptor site, were proposed to account for the rapid sequence divergence of the gene family members. The first mechanism is illustrated by the divergent evolution of the MCE in *SEMG1* and *Svs2*, where short conserved nucleotide sequences at the exon ends have been expanded to yield large exons composed of differently sized repeats. The second mechanism was illustrated by the similarity between rat and mouse *Svs4* and the human semenogelin genes, *SEMG1* and *SEMG2*, where homology was discovered between sequences surroundung the splice acceptor site and the 5' end of MCE in the semenogelin genes and sequences in the intron upstream to MCE in *Svs4*.

A new clue to the origin of the REST genes came by the discovery of their homology with the precursor of the WFDC-type elastase inhibitor elafin, also known as PI3 [19]. Like the REST genes, *PI3* consists of three exons, with the signal peptide encoded by exon 1, and with exon 3 containing 3' non-translated NT only. The second exon of human *PI3* codes for the elastase inhibitor elafin, but also an amino-terminal extension of hexa-peptide repeats [20], which is reported to anchor the inhibitor to the cornified cell envelope in terminally differenti-ated keratinocytes in skin, by way of a transglutaminase (TGase) mediated reaction [21]. The homology between *PI3* and the REST genes was revealed by conserved nucleotide sequences in exons 1 and 3, *i.e.* similar to the conservation between most of the REST genes. Further-more, as many REST gene products and the amino-terminal repeats in the PI3 precursor are reported to be TGase substrates, it was feasible to assume that the REST genes had arosen from a *PI3* related gene by selection for the TGase substrate moiety and subsequently lost the prote-ase inhibitor part by lack of selective pressure.

An elaborated and surprising form of the second mechanism for REST gene evolution was identified in studies on SCP from the guinea pig. It was found that the gene encoding the Svp1 precursor carry sequences homologous with both *SEMG2* and *PI3*, as to suggest a composite gene consisting of semenogelin-like and PI3-like parts [22]. Furthermore, in the large first intron of the gene, there were sequences homologous with both MCE of *SEMG2* and exon 1 of *PI3*, which imply that a splicing event might have joined SPCE in a SEMG2-like gene with the second exon in a neighboring PI3-like gene.

With the advent of DNA sequences from the human genome project, it was possible to pin-point the location of *SEMG1* and *SEMG2* and *PI3* in the genome. It turned out that they were located next to each other at a locus on chromosome 20 in the human genome, lending further support to the hypothesis of a common ancestry. Moreover, this also led to the discovery of a second human WFDC-encoding gene with homology to *SEMG1* and *SEMG2* and eventually to the identification of a larger locus containing the REST genes and related genes with homol-ogy to known or presumed WFDC- and Kunitz-type protease inhibitors [23–25]. In addition to the REST genes *SEMG1* and *SEMG2*, the human locus consists of 17 functional genes con-taining either WFDC or Kunitz sequence motifs, or both. The genes are clustered at two sublo-cus, separated in humans by 215 kb DNA containing unrelated genes. *SEMG1* and *SEMG2* are situated at the sublocus with the more centromeric location on human chromosome 20, together with four genes of proteins with WFDC motifs, two of which, *PI3* and *SLPI*, encodes protease inhibitors with specificity for the elastase. The locus is also conserved in rat and mouse, but in those cases the REST genes are *Svs2-Svs6* [26].

In this report, we have identified two major SCPs, one in rabbit and one in guinea pig, and sequenced their cDNA. We have also investigated the REST gene locus in the mammalian

clade Glires by analysing DNA sequences retrieved from genome databases in order to enhance our knowledge on the origin and evolution of the SCPs.

## Materials and methods

### Samples

Guinea pig (Dunkin-Hartley strain) seminal vesicle tissue and SVS were gifts from Dr Nishtman Dizeyi, Lund University, Department of Clinical Sciences, Malmö. Tissue specimens of reproductive organs from Swedish Lop rabbit were gifts from Dr. Benjamin Pippenger, Straumann, Basel, Switzerland. Samples were stored at -80˚C. Before use SVS samples were weighed, and solubilized and diluted in 40 mM Tris, pH 9.7, 4 M urea, 25 mM EDTA.

**SDS-PAGE.** Analysis by SDS-PAGE was done on 4–20% Mini-protean precast gels (Bio-Rad Laboratories, Hercules, CA, USA) that were run in 25 mM Tris, 192 mM glycine, 0.1% SDS, pH 8.3. Prior to electrophoresis, samples were incubated at 95˚C for 5 min with an equal volume of 2xLaemmli sample buffer (Bio-Rad Laboratories), with or without 0.7 M β-mercaptoethanol. Page ruler (Thermo Fisher Scientific, Waltham, MA, USA) was used as molecular size marker.

**Transglutaminase activity.** TGase treatment of SVS was done with the guinea pig liver enzyme (Sigma-Aldrich, Saint Louis, MO, USA) in the presence of dansylcadaverine (Sigma-Aldrich). The final reaction mix of 20 μl contained SVS diluted more than 100 fold in 50 mM Tris-HCl, pH 7.5, 90 mM NaCl, 5 mM $CaCl_2$, 20 mM DTT, 1 mM dansylcadaverine, and 1.5 mU of TGase. Negative controls run in parallel also included 10 mM EDTA. Samples were incubated at 37˚C for 1 h and then analyzed by SDS-PAGE. Following electrophoresis, the gels were inspected under UV light for dansylcadaverine fluorescence and photographed. Finally the gels were stained for protein with the PageBlue protein stain (Thermo Fisher Scientific).

**Masspectrometric analysis.** In gel enzymatic digestions with trypsin were done on excised protein bands cut into smaller pieces (2 x 2 mm) essentially as described [27, 28]. The tryptic peptides were dissolved in 0.1% formic acid and analysed by nanoflow reversed phase HPLC tandem mass spectrometry using an ESI-LTQ-Orbitrap XL mass spectrometer (Thermo Fisher Scientific) interfaced with an Eksigent 2D NanoLC system (Eksigent Technologies, Dublin, CA, USA). The LC-MSMS analyses were performed as described [27]. The mass spectrometer was operated in data-dependent mode. Each full scan was followed by seven MSMS events using collision-induced dissociation (CID). The generated mass spectra were manually interpreted to identify internal peptide tags with molecular masses and sequences agreeing with the translated virtual tryptic peptide maps of the protein precursors [29, 30].

**Generation of cDNA transcripts.** RNA was isolated from 0.2–0.3 g of tissue specimens homogenized in Trizol (Thermo Fisher Scientific) using a polytron, following the recommendations of the supplier of the reagent. Isolated RNA was solved in nuclease free water and stored at -80˚C. For RT-PCR, samples of 1–2 μg of RNA were taken to cDNA synthesis using the RevertAid RT kit (Thermo Fisher Scientific), after first being freed of contaminating DNA by incubation with 1 U of Dnase I (Invitrogen). Both Dnase treatment and cDNA synthesis, primed with oligo(dT)$_{18}$, were done according to protocols provided with the RevertAid kit. The PCR step was done with Advantage 2 (Clontech Laboratories, Mountain View, CA, USA) in a buffer provided with the polymerase. The reaction volume was 10 μl and included 1 μl of first strand cDNA, 40 mM Tricine-KOH, pH 8.7, 15 mM $KCH_3COO$, 4 μg/ml bovine serum albumin, 0,005% each of Tween 20 and Nonidet-P40, 1 mM dNTP, and 2 μM each of forward and reverse primer. Standard PCR was done with an initial Advantage 2 activation step at 95˚C for 1 min followed by 35 cycles of denaturation at 95˚C for 30 s and annealing/extension at 68˚C for 1 min. The protocol also included a final extension at 68˚C for 1 min. For larger

templates, the annealing/extension time was prolonged with 1 min per expected kb in product size. Sequences of primers are provided as S1 and S2 Tables.

The cDNA synthesis protocol was also used for production of RACE ready cDNA, with the modification that the oligo(dT)$_{18}$ primer was replace by the 3' CDS-primer A (Clontech Laboratories) in 3' RACE and in 5' RACE by custom made oligonucleotides (S1 and S2 Tables). Before synthesis, the reaction yielding 5' RACE ready cDNA was made 1 μM with respect to the SMARTer oligonucleotide (Clontech Laboratories). Following cDNA synthesis, RevertAid was heat inactivated at 70˚C for 10 min and the samples were diluted with 30 μl nuclease free H$_2$O and stored at -20˚C.

**Obtaining full length-cDNA-sequences.** RACE was done according to a protocol provided by Clontech Laboratories. PCR amplification was done with the Advantage 2 polymerase mix, as described for RT-PCR, but with 1 μl of RACE ready cDNA as template and with 0.4 μM Universal primer A mix (Clontech Laboratories), and 0.4 μM custom made gene specific oligonucleotide as primers. A PCR touchdown protocol was used that consisted of an initial Advantage 2 activation step at 94˚C for 1 min, which was followed by 5 cycles of incubation at 94˚C for 30 s and 72˚C for 2 min. Then there was 5 cycles at 94˚C for 30 s, 70˚C for 30 s, and 72˚C for 2 min. Finally there was 27 cycles at 94˚C for 30 s, 68˚C for 30 s, and 72˚C for 2 min.

PCR products were analyzed by electrophoresis in 1% agarose gels run in 89 mM Tris, 89 mM boric acid, 2 mM EDTA, pH 8.3 (TBE buffer) and stained with GelRed (Biotium, Fremont, CA, USA). DNA concentrations were estimated by comparing the staining intensity in agarose gels of PCR products to similarly sized DNA bands of known concentration in Mass-Ruler (Thermo Fisher Scientific). Recovery of PCR products for reamplification was done by punching specific bands on the stained agarose gel with a plastic pipette tip, where after the content was transferred to a tube with 10 μl of TE buffer. 1–3 μl of the recovered material was reamplified according to the protocol described for standard PCR.

Sanger sequencing of DNA was done with 100–400 ng of PCR product and 25 pmol primer at GATC Biotech (Cologne, Germany). Signal peptides were predicted using the SignalP server (http://www.cbs.dtu.dk/services/SignalP).

## Study of the extended REST gene locus in rodents and lagomorphs

Genomic DNA sequences were retrieved from the Gene database at NCBI (https://www.ncbi.nlm.nih.gov/gene) for 14 rodents and 2 lagomorphs (Table 1). Start and stop codons in flanking genes marked the ends of DNA sequences retrieved for a certain genomic loci, *e.g.* for the REST gene loci it was the sequence located between the stop codon of *Matn4* and the start codon of *Kcns1*. Genes at the loci were identified by comparing sequence of the loci to known genes at the human, mouse, and rat loci. Homology search was done by sequence alignment using the computer program BLAST, available at NCBI (https://blast.ncbi.nlm.nih.gov/Blast.cgi). The program also generated dotplots, which were recovered and utilized in the production of figures. If nothing else is stated in legends to figures and supplementary material, the scoring parameters for nucleotide sequence alignments were set as follows: word size: 7, match reward: 1, mismatch penalty: -1, gap penalty: 2, gap extension penalty: 1. For amino acid sequences, the scoring matrix BLOSUM62 was used with conditional compositional scoring matrix adjustment and a gap penalty of 11 and gap extension penalty of 1; the word size was 3. Multiple alignment of protein and DNA sequences were done with the program Clustal Omega, available at (https://www.ebi.ac.uk/Tools/msa/clustalo/), using the default settings.

The loci were also translated in six riding frames and scanned for the signature Cys pattern of WFDC genes.

**Table 1. Animal species selected for analysis of REST gene loci.**

| English name | Scientific name | Designation[a] | Family |
|---|---|---|---|
| House mouse | *Mus musculs* | Mouse | Muridae |
| Brown rat | *Rattus norvegicus* | Rat | Muridae |
| Chinese hamster | *Cricetulus griseus* | Hamster | Cricetidae |
| Prairie deer mouse | *Peromyscus maniculatus* | Deer mouse | Cricetidae |
| Prairie vole | *Microtus ochrogaster* | Vole | Cricetidae |
| Upper Galilee Mountains blind mole-rat | *Nannospalax galili* | UGMBMR | Spalacidae |
| Lesser Egyptian jerboa | *Jaculus jaculus* | Jerboa | Dipodidae |
| Guinea pig | *Cavia porcellus* | | Caviidae |
| Common degu | *Octodon degus* | Degu | Octodontidae |
| Long-tailed chinchilla | *Chinchilla lanigera* | Chinchilla | Chinchillidae |
| Naked mole-rat | *Heterocephalus glaber* | | Heterocephalidae |
| Damaraland mole-rat | *Fukomys damarensis* | | Bathyergidae |
| Ord's kangaroo rat | *Dipodomys ordii* | Kangaroo rat | Heteromyidae |
| Thirteen-lined ground squirrel | *Ictidomys tridecemlineatus* | Ground squirrel | Sciuridae |
| European rabbit | *Oryctolagus cuniculus* | Rabbit | Leporidae |
| American pika | *Ochotona princeps* | Pika | Ochotonidae |

[a]The species name used in the text of this article, when differing from the full English name.

**Nomenclature.** The MGI approved symbols Svs1-Svs6 were used for the mouse REST genes and its orthologs in related species. For the guinea pig REST genes, we used the symbols Svp1-Svp4 for the 4 proteins with different chromatographic properties, which were originally isolated from seminal vesicle secretion [31]. The semen coagulum protein Svp1 is synthesized from a precursor that also generates the two overlapping and almost identical Svs3 and Svs4, in this paper denoted Svs3/4 [9]. In this article we name the gene of this precursor *Svp1*. It is also known under the name GP1G [22].

## Results

### Identification of the major guinea pig and rabbit SCPs

SVS proteins were solubilized and diluted in a high pH buffer containing urea, as previously devised for the human, mouse and rat SCPs. The solubilized material was analyzed by SDS-PAGE and stained for proteins (Fig 1). In the guinea pig SVS sample, Svp1, Svp3/4 and Svp2 were identified by their apparent molecular masses of 25, 20 and 13 kDa. The precursor of Svp1 and Svp3/4, was also detected as a faint band with a molecular mass of 43 kDa. In the upper part of the gel, there was also a previously not reported, heavily stained, protein band with an apparent molecular mass of approximately 190 kDa. This potentially new semen coagulum protein was named Svp5. The rabbit seminal vesicle tissue was filled with a semi-solid mass, assumed to consist of SVS, which was solubilized by the high pH/urea buffer. SDS-PAGE yielded a predominant protein component of 200 kDa, and a few minor, weakly stained, components. The predominant component, presumed to be the major semen coagulum protein in rabbits, was named SVP200. The minor components were not further analyzed. SDS-PAGE was also run without reducing agents in order to detect disulfide-linked multimers. The major components in both the guinea pig and the rabbit samples migrated essentially the same distance, whether reduced or unreduced. From this it was concluded that guinea pig Svp1-Svp5 and rabbit SVP200 do not form disulfide-linked multimers in SVS.

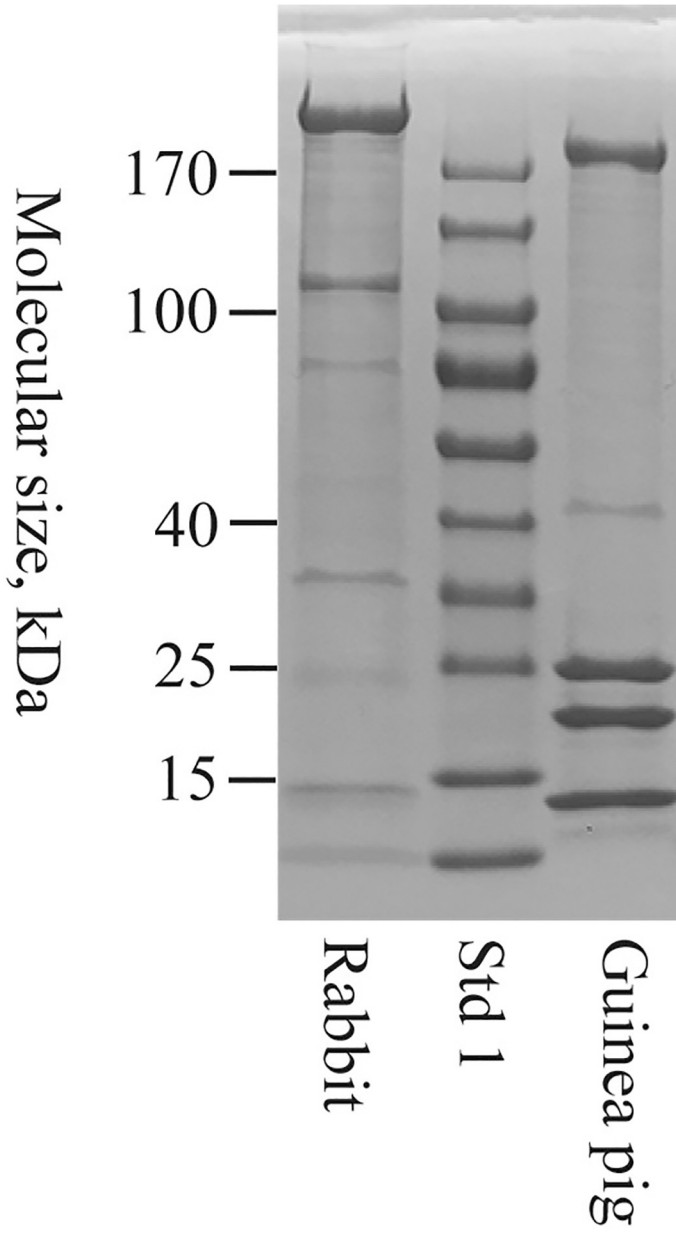

**Fig 1. Analysis of seminal vesicle secretion by SDS-PAGE.** Guinea pig and rabbit sample, equivalent with 0.1–0.2 μl seminal vesicle secretion, were reduced and run on a 4–20% gradient SDS-PAGE gel run in Tris-Glycine buffer, pH 8.3. Std 1 refers to the molecular mass standard Page ruler.

The SCPs are in general excellent substrates of TGase. In order to study the effect of TGase, solubilized samples of SVS were diluted more than 100 fold in reaction buffer and then incubated with TGase in the presence of dansylcadaverine. Following SDS-PAGE, the gel was exposed to UV light and the dansylcadaverine fluorescence was recorded, where after the gels were stained for protein. As can be seen on the gel stained for protein, TGase treatment wiped out the Svp1 and Svp5 components in the guinea pig sample and the SVP200 component in the rabbit sample (Fig 2A). The treatment also generated high molecular mass material with incorporated dansylcadaverine that indicates protein crosslinking by TGase (Fig 2B). Together

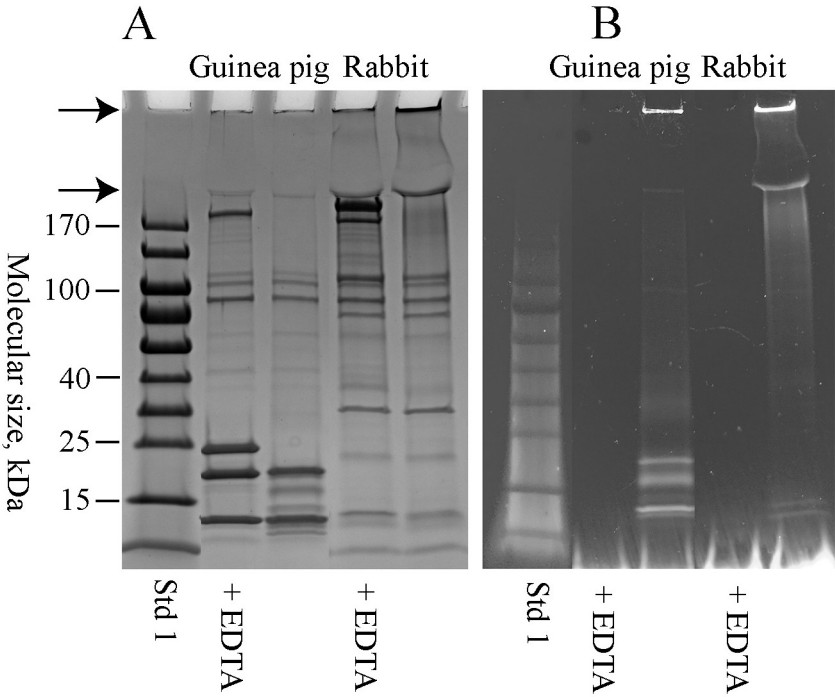

**Fig 2. Transglutaminase treatment of seminal vesicle secretion.** Guinea pig and rabbit samples were incubated with 1.5 mU of transglutaminase for 60 min at 37˚C in the presence of 1 mM dansylcadaverine. Control samples containing EDTA were run in parallel. The samples were reduced and subjected to SDS-PAGE on a 4–20% gradient SDS-PAGE gel run in Tris -glycine buffer, pH 8.3. In (A) the gel is stained for protein and in (B) it is exposed to UV-light in order to visualize the dansylcadaverine fluorescence.

this suggests that Svp1 and Svp5 are the predominant SCPs in the guinea pig and SVP200 in the rabbit.

## Primary structure of the novel guinea pig and rabbit SCPs

The following strategy was used in order to determine transcript and protein primary structures:

1. Potential genes of SCPs were identified in genome databases

2. cDNA transcripts were generated by RT-PCR, using RNA isolated from seminal vesicle tissue and oligonucleotides based on nucleotide sequences in genes of potential SCPs.

3. Transcript sequences were determined and then extended by RACE, in order to obtain the fullength cDNA sequence

4. Nucleotide sequences of transcripts were translated and used to produce virtual peptide maps that were compared to peptides maps generated by mass spectrometry on tryptic digests of isolated protein bands from the SDS-PAGE gel. In this way, the conformity of transcripts and proteins were made

**Guinea pig Svp5.** Previous studies on rodents and primates have shown that the genes of semen coagulum proteins are almost exclusively located between the genes *SLPI* and *WFDC5*. Nucleotide sequences bridging these two genes in the guinea pig genome were probed by sequences of human *SEMG2* and *PI3* using BLAST. There were 4 regions with homology to

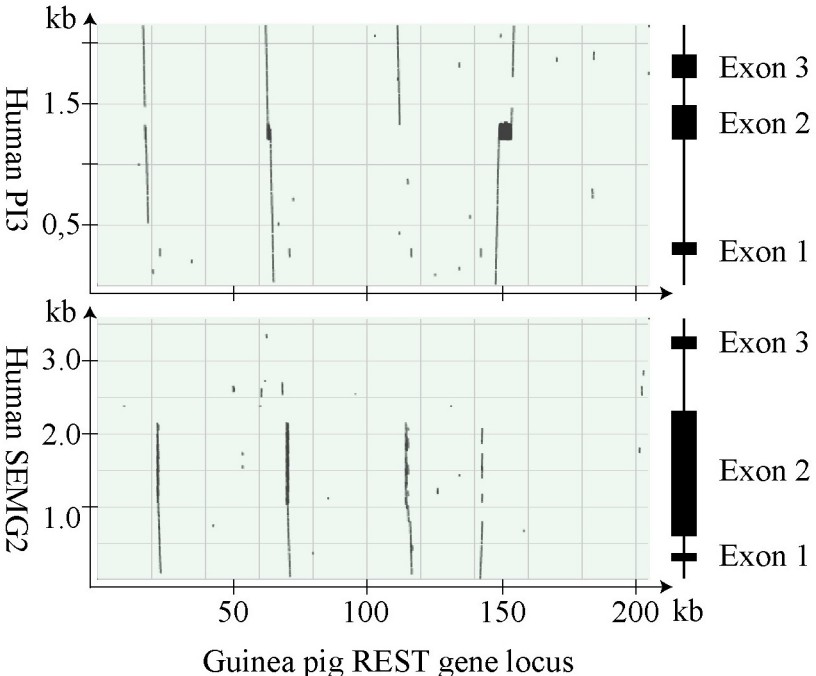

**Fig 3. Dotplots showing sequence homology with *SEMG2* and *PI3* at the guinea pig REST gene locus.** The computer program BLAST was used to align guinea pig genomic sequences, located between *Slpi* and *Wfdc5* at the REST gene locus, and the nucleotide sequences of human *SEMG2* and *PI3*. The result is displayed in dotplots with illustrations to the right depicting genes with the location of exons indicated by boxes.

both *SEMG2* and *PI3* in the sequence of 204 kb that separated guinea pig *Slpi* and *Wfdc5* (Fig 3).

In three of the regions we identified *Svp2*, *Svp1*, and a *PI3*-like gene, encoding a protein denoted caltrin II. The fourth region carried a probable gene containing a long open reading frame that to a large extent consisted of imperfect 72 bp tandem repeats. RT-PCR showed that the gene is transcribed in the guinea pig seminal vesicles. The sequence of the fullength transcript was determined and consisted of 5,106 NT plus a poly-A tail of undetermined size. The transcript could be translated to a protein precursor of 1,557AA that yielded a virtual tryptic peptide map in which 28 peptides could be identified by mass spectrometry on trypsin digested Svp5, based on their molecular masses and AA sequences (S1 Fig). The peptides covered most of the protein precursor, from position 30 to position 1541. Homology with the SEMG2 precursor suggested that the Svp5 precursor carried a signal peptide of 21 AA, something that was also predicted by analysis on the signalP server. The mature secreted protein would then consist of 1,536 AA with a molecular mass of 161 kDa. It had an unusually high pI of 10.3, similar to the major SCPs in human, mouse, and rat. The primary structure had a very high content of Gly, Lys, Val, Gln, and Leu, which together constituted 56% of the total number of AA. Much of the structure was arranged in tandem repeats with a basic repeat length of 24 AA, which could also be part of lager tandem repeats. Characteristic to the primary structure was also two types of three residue motifs Val-Lys-Gly and Gly-Gln-Asp. A comparison with human PI3 shows that the repeats were homologous with the repeats of the TGase substrate domain (TSD) located upstream to the protease inhibitor domain of the PI3 precursor (Fig 4). At the C-terminus, the peptide chain carried the motif of a WFDC domain with homology to PI3. The molecular size calculated from the primary structure is 15% smaller than the size

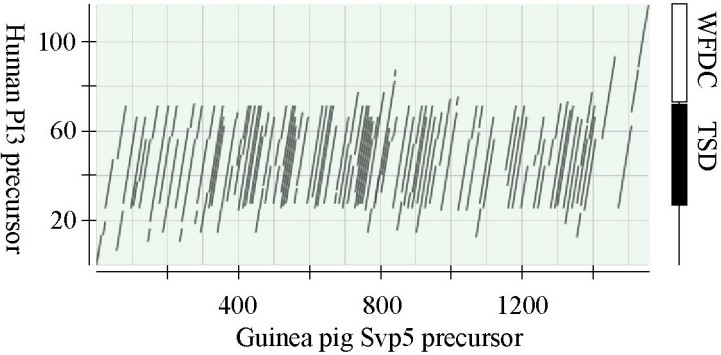

**Fig 4. Homology between Svp5 and the TSD of PI3.** The amino acid sequences of guinea pig Svp5 and human PI3 precursors were aligned using the BLAST program. The result is displayed in a dotplot. The illustrations to left depict the location of the WFDC and the transglutaminase substrate domains in PI3.

estimated by SDS-PAGE. The peptide chain carried no signals for N-linked glycosylation, indicating that the size difference is not due to glycosylation as has been reported for human SEMG2.

The *Svp5* transcript was also compared to the guinea pig genomic sequence in the Gene database at NCBI. This demonstrated that *Svp5* consists of 2 exons, with the first exon of 94 bp carrying 18 bp of 5' non-translated NT, 63 bp encoding the signal peptide, and 13 bp coding for the immediate N-terminus of the secreted protein. The second exon of 5,018 bp holds the remaining 4,595 bp coding for Svp5, which is followed by the stop codon and 420 bp 3' non translated NT with a poly-adenylation signal located 15–20 bp upstream to the end of the exon.

**Homology of guinea pig *Svp5* and human *SEMG2* and *PI3*.**   The upstream promoter region, the first exon, and the beginning of the intron showed homology with human *SEMG2*. However, the homology in the intron was not confined to the first intron of *SEMG2*, as it was also homologous with MCE sequences, *i.e.* sequences that are translated in the generation of secreted SEMG2 are non-translated and located in the intron of Svp5 (Fig 5). Further downstream in the *Svp5* intron, there were sequences homologous with the upstream promoter region, the first exon, and the first intron of *PI3*. The second exon of *Svp5* was homologous with the second exon of *PI3*, but with a huge expansion in the number of repeats homologous with those that generates the TSD in PI3. Sequences homologous with exon 3 in PI3, were found in flanking sequences 3' to *Svp5*. Because of the conserved sequences, *Svp5* could be regarded as a composite gene, which presumably was created by the merger of a *SEMG2*- and a *PI3*-like gene. We also found that there were 3 differences between the *Svp5* transcript and the genomic sequence in the database, two of which generated AA replacements and a third that yielded a shift of reading frame immediately upstream to the WFDC motif of the gene (S2 Fig). As a consequence of a gap in the genomic sequence, the gene sequence encoding the C-terminus, the stop codon, and 233 bp of 3' non translated NT were not available in the database.

In a second data base search, Svp5 was found to be identical with a previously described protein, denoted guinea pig trappin, which had been identified because of homology with the PI3-like protein caltrin II. The sequences of the *Svp5* and trappin transcripts differed in 7 positions, two of which were neutral and five that generated amino acid substitutions (S2 Fig).

**Rabbit SVP200.**   The size of the genomic region separating rabbit *SLPI* and *WFDC5* was 84 kb. It contained a likely pseudogene with homology to human *SEMG2*, but also three

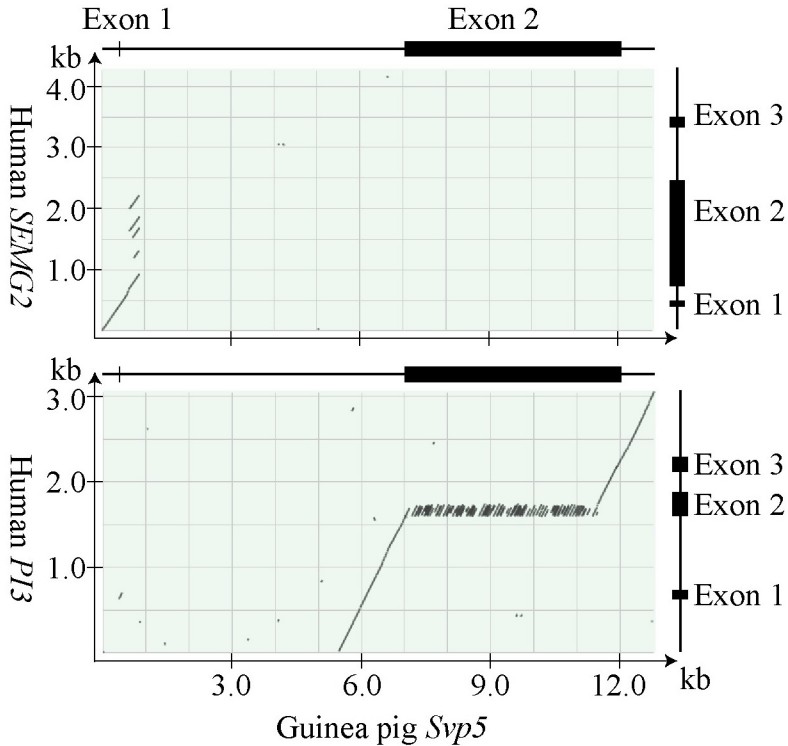

**Fig 5. Homology of guinea pig *Svp5* and human *SEMG2* and *PI3*.** The dotplots are made with the results of BLAST alignments, showing homology with *SEMG2* in the 5' end and *PI3* in the 3' end of *Svp5*. The lines at the top and to the right of the dotplots illustrate genes, with boxes depicting the location of exons.

regions encompassing putative composite genes, displaying non-overlapping sequence similarity with *SEMG2* and *PI3*, with the putative 5' regulatory and signal peptide-encoding NT showing homology with *SEMG2*, while the reminder of the coding NT and 3' non-translated NT were homologous with *PI3* (Fig 6). One of these genes carried mutations indicative of a pseudogene, whereas a second gene was potentially functional and showed similarity to human PI3 and to some extent also to the gene encoding caltrin II in the guinea pig. The third gene carried tandem repeats of 72 bp, similar to guinea pig *Svp5*. RT-PCR demonstrated that the transcript of this gene was abundant in rabbit seminal vesicle tissue, suggesting that it might be the gene encoding SVP200. It was therefore taken for further analysis.

The fullength rabbit seminal vesicle transcript was found to be 3,853 NT plus a poly-A tail of undetermined size. It could be translated to a polypeptide chain of 1,170 AA, including 20 residues at the N-terminus that probably formed a signal peptide, as indicated by the similarity with analogous structure in primate and rodent SCPs and as predicted by the SignalP server. Mass spectrometry identified 26 peptides in tryptic digests of SVP200 agreeing in size and sequence with peptides predicted from the deduced primary structure of SVP200 (S3 Fig). The calculated molecular mass of the mature SVP200 was 126 kDa and it had a pI of 4.9. There were no signals for N-linked glycosylation in the peptide chain, suggesting that SVP200 probably was not glycosylated. The polypeptide chain consisted to a large extent of poorly conserved tandem repeats, in which the predominant repeat units was 24 aa, but there were also sequences that seemed to be derivatives and multiples of this repeat unit. A molecular comparison showed that much of the primary structure of SVP200 was homologous with the TSD of human PI3, similar to guinea pig Svp5, but was lacking the WFDC domain situated at the C-

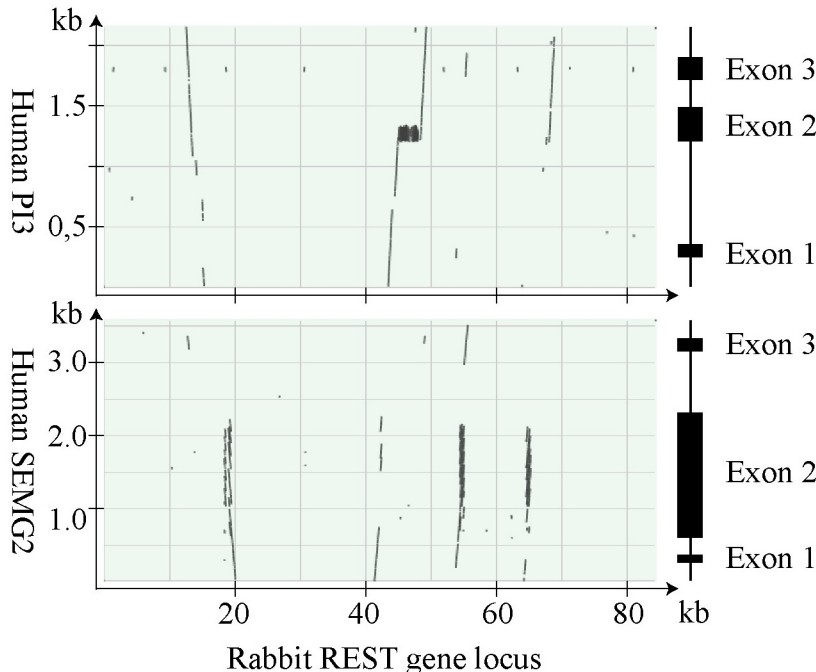

**Fig 6. Dotplot showing sequence homology with *SEMG2* and *PI3* at the rabbit REST gene locus.** The computer program BLAST was used to align rabbit genomic sequences, located between *SLP* and *WFDC5* at the REST gene locus, and the nucleotide sequences of human *SEMG2* and *PI3*. The lines with boxes to the right illustrate the genes with boxes depicting the location of exons.

terminus of both PI3 and Svp5. Furthermore, the core of SVP200 and Svp5 seemed to be similar in structure, as BLAST analysis demonstrated that the primary structure at position 132–1097 of the SVP200 precursor was similar to position 251–1119 of the guinea pig Svp5 precursor with on average 35% conserved residues. As in guinea pig Svp5, the peptide chain of SVP200 also had a high content of Gln, Lys and Val, whereas the high Gly and Leu content in Svp5 was replaced by a high content of Pro and Glu. The size discrepancy in molecular mass calculated from the amino acid composition and the one estimated by SDS-PAGE, suggested that SVP200 might possess unusual molecular properties leading to reduced mobility on SDS-PAGE.

The nucleotide sequence of the transcript was compared to the genomic DNA sequence in the Rabbit Genome database. This showed that the gene consisted of three exons with a slightly different organization than in previously analyzed genes of SCPs. The first exon of 91 bp carried 18 bp of 5' non-translated NT, 60 bp encoding the signal peptide and 13 bp yielding the amino terminus of the secreted SVP200. The size of the second exon was 168 bp and coded for what might be a unique N-terminal domain that carried 4 Trp—a residue that was absent from the remaining poly peptide chain. Interestingly, NT homologous with this exon were found in the first intron of both human *PI3* and in guinea pig *Svp5*. The third exon of 3,594 bp is the MCE and held 3,269 bp coding for the rest of the protein, carried the stop codon, and 322 bp of 3' non-translated NT. The analysis also showed a substantial deviation both in size and structure of the *SVP200* transcript from the one predicted on the basis of the gene sequence in the database, *e.g.* it was 144 bp longer than expected from the genomic sequence and there were also several sequence differences, most of them located in a region from approximately nucleotide position 1780–2760 (S4 Fig).

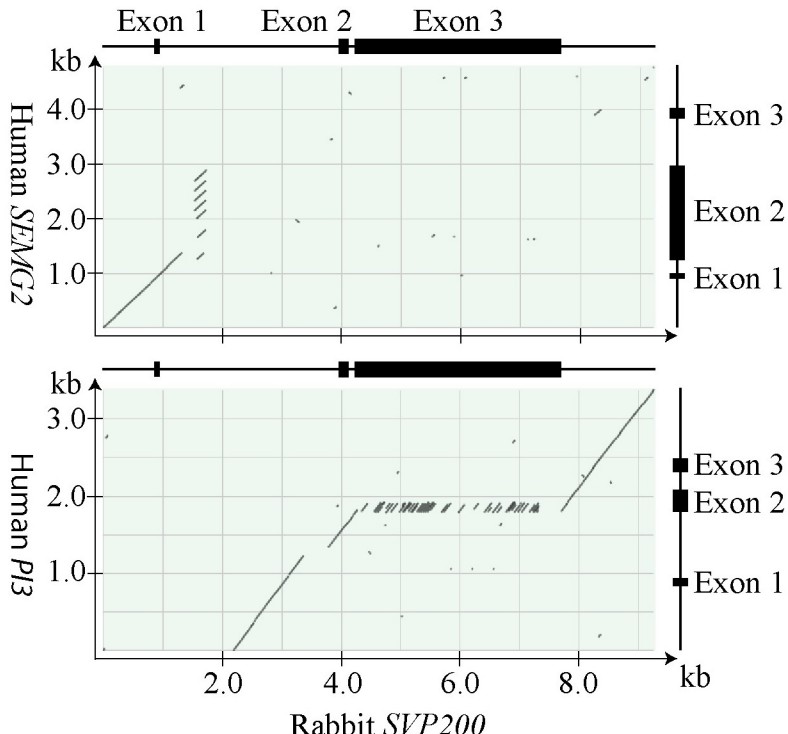

**Fig 7. Homology of rabbit *SVP200* and human *SEMG2* and *PI3*.** The dotplots illustrate the results of BLAST alignments, showing homology with *SEMG2* in the 5' end and *PI3* in the 3' end of *SVP200*. The lines at the top and to the right of the dotplots illustrate genes, with boxes depicting the location of exons.

**Homology of rabbit *SVP200* and human *SEMG2* and *PI3*.** A sequence comparison of rabbit *SVP200* and human *SEMG2* and *PI3* was made and analyzed in detail. The result, displayed in dotplots, demonstrated that there was a continuous sequence, encompassing the upstream promoter region, exon 1, and the 5' end of the first intron in *SVP200*, which was homologous with human SEMG2. The conserved sequence in the first intron included NT that were homologous with the first intron, but also of coding NT in the 5' end of the MCE in *SEMG2* (Fig 7). The comparison with human *PI3* identified sequences with homology to all three exons and much of the intron sequences of *PI3* (Fig 7). The sequence with homology to exon 1 of *PI3* was located in the first intron of *SVP200*, but the presence of deleterious mutations suggested that it was a pseudo exon, which could not be used as an alternative first exon. Most of the MCE consisted of repeats that were homologous with NT that code for the TSD of PI3, as pointed out above. Immediately downstream of the polyadenylation site in *SVP200*, there was a continuous nucleotide sequence with homology to most of exon 2, intron 2, exon 3, and 3' flanking sequence of *PI3* (S5 Fig).

## The extended REST gene locus in rodents and lagomorphs

The DNA sequences of genetic loci encompassing REST genes and related genes encoding WFDC motifs were retrieved from genome databases of rodents and lagomorphs. The locus was defined as the nucleotide sequence located between the conserved flanking genes *Matn4* and *Kcns1*. It displayed a relatively large size variation between species, as exemplified by the locus size of 68 kb in the pika and 3,020 kb in the guinea pig. Conserved genes were identified by pairwise sequence alignment of genomic regions and individual human, rat, and guinea pig

REST and WFDC genes. This demonstrated that, in addition to the size variation, there were also big differences in the number of genes at the locus, ranging from 3 in the pika to 14 in the rat (Fig 8). As can also be seen, the rodent suborders myomorpha and hystricomorpha are exemplified by species from more than one taxonomic family, whereas castorimorph and sciuromorph rodents are represented with only one species each. There was no genomic sequence available for a species of the anomaluomorpha suborder. The lagomorphs were represented by rabbit and pika, *i.e.* one species each of the extant families Leporidae and Ochotonidae.

In the four analyzed rodent suborders, the REST genes seemed to fall into 4 categories. The myomorph rodents carried 6 REST genes with close similarity to mouse *Svs2-Svs6*; the hystricomorph rodents carried 3 REST genes with close similarity to guinea pig *Svp1*, *Svp2*, and *Svp5*; in the kangaroo rat, a castorimorph rodent, we identified 4 previously not described REST genes based on their similarity to human *SEMG2*; the ground squirrel, a sciuromorph rodent, carried a pseudogene homologous with human *SEMG2*, but no functional REST gene. In the lagomorph suborder, the pika was devoid of REST genes, whereas the rabbit carried two pseudogenes and the functional REST gene *SVP200*.

## Myomorph REST genes

A full complement of 6 REST genes was identified in the genomic sequence from 6 of the analyzed myomorph species. The DNA sequences of *Svs5*, *Svs6* and *Svs3a* were not available from a 7th species, the jerboa, because the sequence of the genomic region where theses gene were expected to be found was missing in the database. A further analysis demonstrated that several of the myomorph REST genes were affected by mutations, suggesting that they were either truncated functional genes or non-functional pseudogenes (Fig 8).

**Homology with *SEMG2* and *PI3*.**  Sequence comparisons with human *SEMG2* and *PI3* revealed only limited conservation of *PI3* sequences in myomorph REST genes, as exemplified by the conserved DNA sequences confined to SPCE and 3NTE in the rat (S6 and S7 Figs). In contrast, *SEMG2* showed extensive similarity to all myomorph REST genes. Dotplots demonstrated that the upstream promoter region, SPCE, intron 1, much of intron 2, and 3NTE of human *SEMG2* were conserved in all myomorph REST genes (S8–S12 Figs). A closer inspection of the aligned sequences showed that all myomorph REST genes also carried sequences with homology to the 5' and 3' ends of the MCE in *SEMG2*. Furthermore, the splice acceptor site 5' to MCE was also conserved in *Svs2*, *Svs3a*, and *Svs3b* (S13 and S14 Figs) This was not the case in *Svs4*, *Svs5* and *Svs6*, where the splice acceptor site was found approximately 0.2 kb further downstream (S15–S17 Figs). Thus, nucleotide sequences that were homologous with coding NT in *SEMG2* were found to be located in the first intron of *Svs4-Svs6*. The splice donor site and sequences surrounding the 3' end of MCE in *SEMG2* were conserved in all of the analyzed myomorph REST genes (S13–S17 Figs).

**Comparison of myomorph REST genes within a species and between closely related species.**  Pairwise comparison of myomorph REST genes demonstrated that the two Svs3 genes, *i.e. Svs3a* and *Svs3b*, were almost identical within any of the analyzed species of murid and cricetid rodents, with 95% or more conserved NT. In contrast, the orthologous forms of *Svs3a* and *Svs3b* were found to differ substantially, *e.g.* there were only around 80% conserved NT in *Svs3a* from closely related species like rat and mouse or deer mouse and vole. It was also found that the two Svs3 genes and *Svs2* were more closely related to each other than they were to *Svs4-Svs6*. In dot plots generated by comparing *Svs3a* or *Svs3b* with *Svs2*, it was found that also much of the MCE was conserved and that a major reason for the difference in the exon sequences was due to short tandem repeats (Fig 9). As can be seen, in murids and cricetides

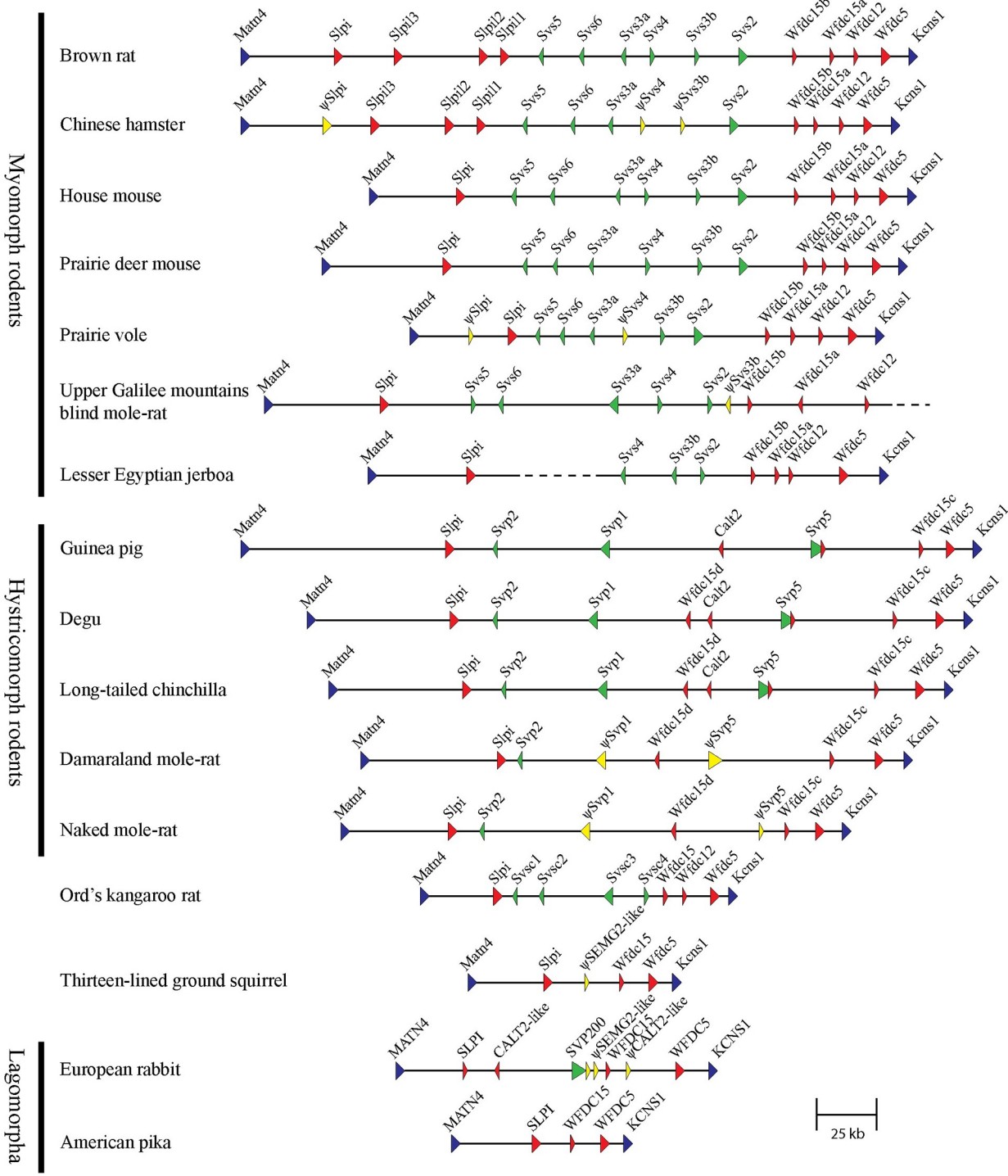

**Fig 8. Outline of the REST gene loci in Glires.** The loci are illustrated as horizontal lines, with arrowheads showing the approximate location and orientation of genes. The colors depict the genes as follows: blue for locus flanking genes, red for WFDC genes, green for REST genes, yellow for pseudogenes, and grey for genes with uncertain status (*i.e.* they might be truncated functional genes or pseudogenes). Genes encoding one or two WFDC domains are indicated by small and large arrowheads respectively. The arrowheads depicting REST genes are small, medium, or large in order to reflect the relative size of their transcripts.

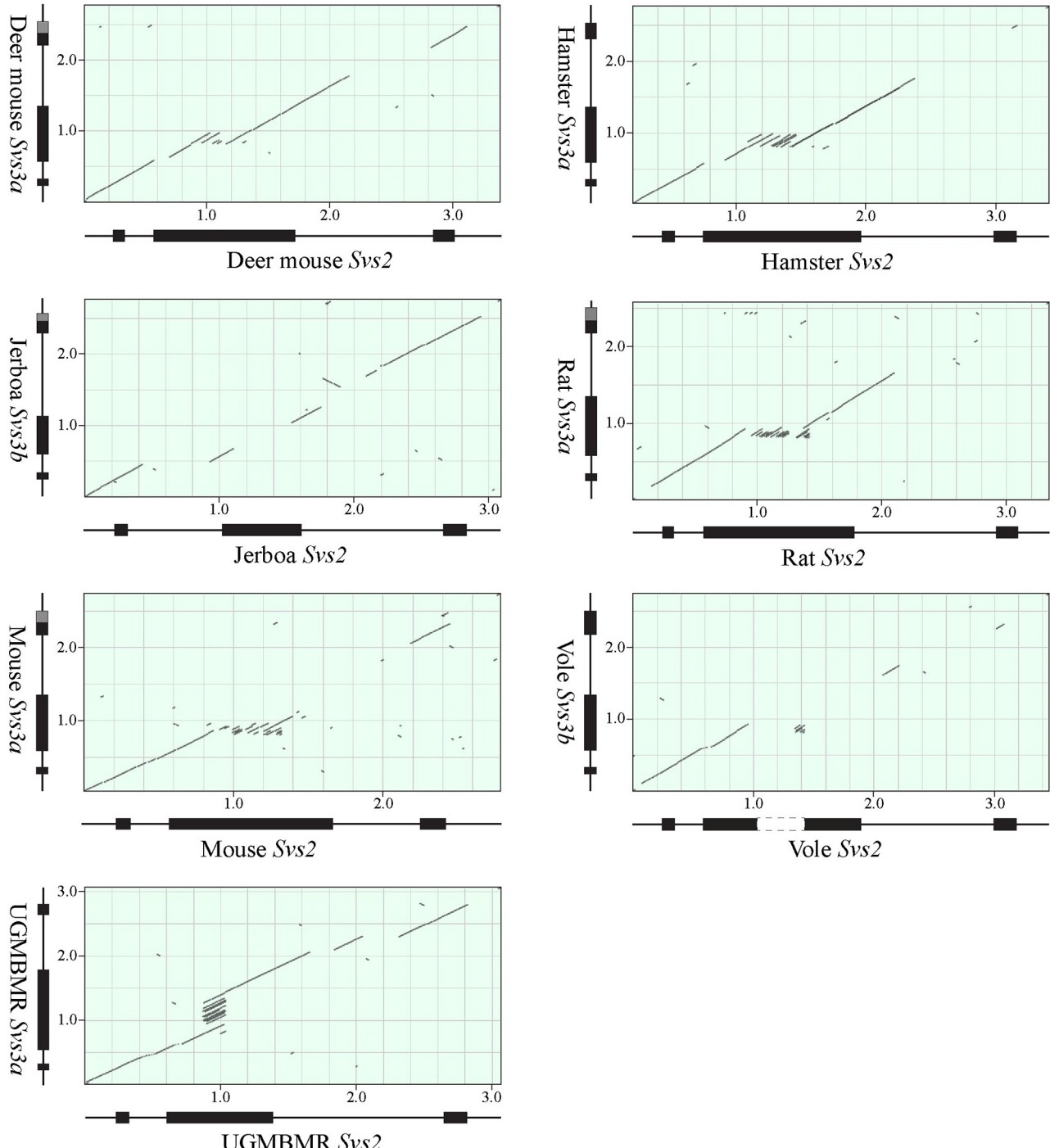

**Fig 9. Homology of *Svs2* and *Svs3* in myomorph rodents.** Dotplots were generated by aligning sequences with the computor program BLAST. The drawings diplayed under and to the right of the dotplots are schematic illustrations of the genes with exons shown as boxes. A gap in the genomic sequence of vole Svs2 is painted white. The dark grey parts in exon 3 of Svs3 depict that there are alternative poly-adenylation sites within the displayed gene or that Svs3a and Svs3b have different poly-adenylation sites. The numbers on the axis show distance in kb from the beginning of the aligned sequences. Note the tandem repeats in the central part of exon 2 in deer mouse, hamster, rat, and muse *Svs2*, as well as the equally positioned repeats in UGMBMR *Svs3*.

these repeats were found in *Svs2*, but in the Upper Galilee Mountains blind mole-rat (UGMBMR), a spalacid species, the short tandem repeats were found in *Svs3a*.

The similarity in transcript size and conserved splice acceptor site in *Svs4-Svs6* indicated that these genes were more closely related to each other, than they were to *Svs2*, *Svs3a* and *Svs3b*. The relationship was further strengthened by BLAST alignments of *Svs4* with *Svs5* and *Svs6* in the deer mouse and the UGMBMR, which showed that the homology of the small Svs gens comprised the whole genes, including MCE, something that was not fully evident when only the rat and mouse genes were compared (S18 Fig). It was also found that the 5' half of *Svs5* and *Svs6*, encompassing the upstream promoter region, SPCE, intron 1, and the beginning of MCE, were structurally very similar within any of the murid and cricetid species with over 90% sequence conservation, whereas this region in the orthologous forms of *Svs5* and *Svs6* were only around 80% similar in sequence. The 3' half of *Svs5* and *Svs6*, containing the major part of MCE, intron 2 and 3NTE, was less conserved with around 60% identically placed NT.

The predicted primary structure of protein precursors of the myomorph REST genes were aligned using the Clustal Omega algorithm. The sequences did not align well due to size heterogeneity and a generally low conservation of the central parts of orthologous proteins in the analyzed species. The size heterogeneity was most pronounced in Svs2, where even closely related species in the families Muridae and Cricetidae, displayed differences in size due to the length of a region encompassing poorly conserved tandem repeats at the center of the molecule (S19 Fig). The repeat region was preceded by a conserved Cys in all analyzed *Svs2* of myomorph species. The poorly conserved repeats could perhaps best be described as consisting of a core with a single hydrophobic residue, frequently a Leu, located between a Gln and a Lys, which in turn were flanked by 3 or 4 residues, in many cases carrying small side chains, but that at regular intervals also contained a residue with an aromatic side chain. In addition to the repeats at the center of the molecules, members of the Cricetidae family, *i.e.* hamster, vole, and deer mouse, carried tandem repeats of 4 or 5 residues close to the amino terminus of the secreted Svs2. The Svs2 molecules of jerboa and UGMBMR were lacking the extended repeat region at the center of the molecules and were therefore substantially smaller than Svs2 from the murid and cricetid species. However, there was a duplication of 14 AA in the amino-terminal part of UGMBMR Svs2, which was not present in the other analyzed myomorph species.

The two isoforms of Svs3, *i.e.* Svs3a and Svs3b, displayed structural similarity to Svs2. In murids and cricetids this was particularly evident in the N-terminal half of the molecule, with close to 50% fully conserved AA, including the single Cys located prior to the central repeat region of Svs2. The C-terminal half was less similar to Svs2, with only around 30% fully conserved residues. This region also carried two Cys, located 17 and 15 residues upstream to the C terminus, which was characteristic to most of the analyzed Svs3 molecules. However, in Jerboa Svs3b there was a single C-terminal Cys located 13 AA upstream from the end of the peptide chain. There was no extended repeat region at the center of murid and cricetid Svs3 that generated size differences as in Svs2 (S20 Fig). Instead, this was found in the UGMBMR Svs3a, where the central part was expanded in a similar fashion as in murid and cricetid Svs2, yielding a molecule that was twice the size of Svs2 in this species (S21 Fig).

The small REST genes *Svs4-Svs6* were found to have some common features, like size and sequence heterogeneity of orthologous protein products, and inability to form disulfide linked complexes due to lack of Cys in the primary structure. They also carried signal peptides that were similar in both size and structure. However, mouse and rat *Svs6* carry a small intron at a position, which in *Svs4* and *Svs5* would be located in MCE. This separates the coding information of mouse and rat *Svs6* into three exons, instead of two, as in *Svs2-Svs5*. The extra *Svs6* intron is most likely absent in the other analyzed myomorph rodents as appropriately located

splice donor or acceptor sites are missing in their genes (S22 Fig). The splice donor site downstream of exon 2 is mutated or deleted in hamster and deer mouse and the splice acceptor site preceding exon 3 is mutated in the hamster, vole and UGMBMR. However, the reading frame is not affected and the location of the stop codon is conserved in all of the analyzed genes, except for *Svs6* of UGMBMR, where a mutation leading to a shift of reading frame generated three unique C-terminal AA and premature stop 4 NT prior the preferred stop in the other genes.

The aligned sequences of Svs4, Svs5, and Svs6 demonstrated conserved AA, especially in the signal peptides generated by SPCE (S23 Fig). The remaining primary structures, emanating primarily from MCE, were less conserved, but still held around 25% highly conserved residues. The fraction of conserved residues were higher in both the N- and C-terminal parts of the molecules, than in the central part, which was also affected by gaps in the aligned sequences. The vole Svs6 was unique among the small Svs molecules by containing a stretch of 6 central tandem repeats (S23 Fig). Unlike the poorly conserved repeats in Svs2, the 6 tandem repeats in vole Svs6 consisted of 18 highly conserved AA.

## Hystricomorph REST genes

The genomes of analyzed hystricomorph rodents had 3 homologous REST genes, *Svp1*, *Svp2*, and *Svp5*, with close similarity to human *SEMG2* and *PI3* (S24–S26 Figs). The upstream promotor region, SPCE, and the 5' end of the first intron were homologous with human *SEMG2*. Furthermore, around 0.5 kb of the sequence in the first intron was homologous with coding NT in MCE of *SEMG2*. Spanning from the 3' half of the first intron to sequences flanking the 3' end of the gene, virtually all hystricomorph REST genes were homologous with the entire *PI3*, including gene flanking sequences. However, in the naked mole-rat most of *Svp5* was deleted, suggesting that it was a pseudogene. Also in Damaraland mole-rat *Svp5* was most likely a pseudogene, as the MCE carried a mutated splice acceptor site and the gene also carried several frame shift mutations and stop codons. Sequence analyses indicated that *Svp1* and *Svp2* consisted of three exons in all analyzed species, similar to human *SEMG2* and *PI3*. This was not the case with *Svp5*, which in the guinea pig was composed of two exons, as the splice donor site downstream of MCE was mutated and therefore retained NT homologous with intron 2 in human *PI3* in the 3' non-translated region. Also degu *Svp5* carried mutations that might prevent spicing out of a second intron, whereas in the chinchilla both the splice donor and acceptor site of MCE seemed to be conserved (S27 Fig). A relatively large first intron, making up around half of the gene in *Svp5* and up to 85% in *Svp1* and *Svp2*, was characteristic to the hystricomorph REST genes. The intron contained sequences homologous with coding NT in both *SEMG2* and *PI3*.

Sequence alignment of the translation products demonstrated that Svp2 is relatively well conserved with over 60% identically placed residues in the 5 analyzed species (S28 Fig). The similarity in primary structure between Svp2 and PI3 is very low in spite of the conserved gene structure, which included conserved intron and exon sequences, and preserved splice donor and acceptor sites (S28 Fig). The defining subunits of PI3, a TSD rich in Gln and Lys, and a Cys-rich WFDC domain, were absent in Svp2.

The AA sequence of approximately 30 residues at the amino terminus of the secreted Svp1 precursor was similar to the amino terminal sequence of secreted Svp2. In the Svp1 precursor this was followed by a region consisting of two types of 24 AA tandem repeats. The repeats located closer to the amino terminus continued to a conserved sequence at the site where the guinea pig Svp1 precursor is cleaved to generate Svp3/4 and Svp1 (S29 Fig). As can be seen, this type of repeat was almost devoid of Gln, a residue essential for TGase cross-linking. The

number of repeats also varied in the different species, suggesting that the size of Svp3/4 varied between species. This was also true for the repeats located C-terminal to the processing site, *i.e.* in Svp1. Both the repeats of the Svs3/4 and the Svs1 domain displayed homology with repeats of the TSD in PI3, but only Svs1 contained multiple Gln and Lys suitable for protein cross linking. As can be seen in the S29 Fig, both naked mole-rat and Damaraland mole-rat carry multiple stop codons, suggesting that *Svp1* in these species might be psudogenes. However, the first stop codon in Damaraland mole-rat is located at the site of proteolytic processing in the precusor molecule, which might indicate that *Svp1* of Damarland mole-rat could be functional to generate Svp3/4, but not Svp1.

The large number of tandem repeats related to the TSD in PI3 and a high proportion of Gln and Lys was characteristic to the analyzed Svp5 molecules. A majority of the repeats could, similar to those of Svp1, best be described as consisting of 24 AA, but multiples or derivatives of that size was also present. Common with Svp1 was also the frequent occurrence of the tripeptide sequence Lys-Gly-Gln in the primary structure. Furthermore, the number of repeats varied considerably between species. The N-terminal part, encompassing the signal sequence, the amino teminus of the secreted protein and a few repeats aligned fairly well, as did the very C-teminal part (S30 Fig). Many of the repeats in the central part of the molecules were highly similar between species, but they were scattered in a way that prevented meaningful alignments. At the C-terminal end of Svp5 there was a structure with homology to the WFDC domain of PI3 in all of the analyzed species. However, the conserved residues of PI3 and the related SLPI, involved in elastase binding, were mutated in Svp5.

## REST genes of the kangaroo rat

Four putative REST genes were identified in the kangaroo rat by their similarity to human *SEMG2* (Fig 10). As can also be seen, the genes showed very little similarity to human *PI3* and were in that respect more reminiscent of the REST genes in myomorph rodents than to those of the rabbit and the hystricomorph rodents.

The genes, tentatively denoted *Svsc1-Svsc4*, also displayed the same organisation with SPCE, MCE and 3NTE, as most other REST genes (S31–S34 Figs). This was also confirmed by sequence alignment with *Svs2* and *Svs4*, which demonstrated that sequences in SPCE and 3NTE of these genes were highly conserved in the four kangaroo rat REST genes. It also demonstrated that sequences surrounding the splice acceptor site and a few coding NT at the 5' end of MCE in *Svs2* were conserved in the kangaroo rat REST genes, but not much of the remaining parts of the exon, except for the splice donor sit at the 3' end. None of the kangaroo rat REST genes used a MCE splice acceptor site that was homologous with that of *Svs4*. Analysis of the translation products showed that Svsc1 and Svsc2 had relatively few Gln in the primary structure, whereas this residue was relatively abundant in Svsc3 and Svsc4. In Svsc3 they also formed poly-Gln tracts; one consisting of 12 residues in the amino terminal part of the molecule and one of 6 residues close to the carboxy terminus. In between these parts was a region consisting of tandem repeats of 12 AA, which formed more than half of the secreted molecule. The repeat region was preceded by a Cys, which was conserved in Svsc4 and also showed homology with the Cys preceding the repeat region in Svs2 and Svs3 of myomorph rodents. Most repeats also carried the Gln-Leu-Lys sequences, which was also found in the repeat regions of myomorph Svs2 and Svs3. Pairwise alignment of the kangaroo rat Svsc1-4 precursors showed that the predicted signal peptide and a few residues at the N-terminus of the secreted protein, encoded by the SPCE, were conserved in all of them. In the rest of the secreted proteins there were similarity in the sequences of Svsc1 and Svsc2, particularly in the N- and C-terminal parts of the peptide chains, whereas the central part was less conserved

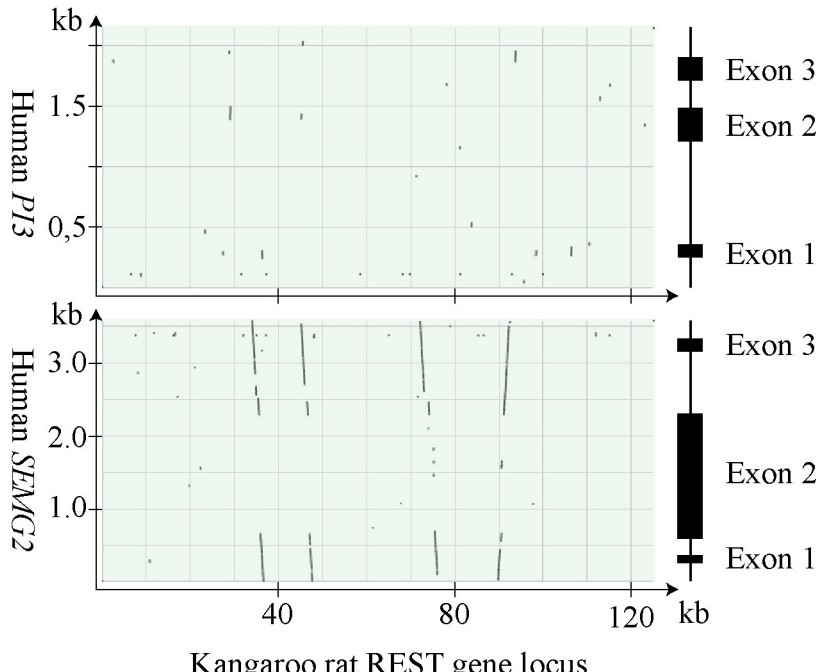

**Fig 10. Homology with human *SEMG2* at the REST gene locus in the kangaroo rat.** The Kangaroo rat's REST gene locus was searched for DNA sequences showing homology with human SEMG2 and PI3 using BLAST. The result displayed as dotplots are shown with illustrations of the human genes to the right.

(S35 Fig). We also found conserved sequences in the N-terminal parts of secreted Svsc3 and Svsc4, but a few AA downstream of the conserved Cys mentioned above, the homology expired and was followed by the tandem repeats in Svsc3. Following the repeat region, the sequence similarity between Svsc3 and Svsc4 resumed for around 30 residues. In Svsc4 this sequence contained a stretch of 25 AA, with a 24 AA tandem copy located downstream. Characteristic to the C-terminal part of Svsc4 were 8 Lys-Gly di-peptides, 6 of which were located in the tandem repeats. The C-terminal part of Svsc3 carried 2 Cys separated by an Arg, with an almost identical location as the Cys-Tyr-Cys sequence that was found close to the C-terminus of Svs3.

## The non-REST genes

In pica and ground squirrel, *i.e.* species lacking functional REST genes, the locus was small and carried 3 genes coding for WFDC domains: *Slpi*, *Wfdc15*, and *Wfdc5*. These three WFDC genes were also present in the reminder of the analyzed species, but in some of them *Slpi* and/or *Wfdc15* were duplicated (Fig 8). *Wfdc12* was present in the myomorph rodents and Ord's kangaroo rat, but not in hystricomorph rodents and lagomorpha. There were also genes with relatively close similarity to *PI3* in the hystricomorph rodents and the rabbit, but not in the other analyzed animals.

   *Slpi* had generated 3 additional copies in the rat and the hamster and 1 extra copy in the vole. In the latter two species, the gene copy closest to the locus flanking *Matn4* was probably non-functional, due to premature stop or mutated translation initiation codons. Lacking similarity with the protease-interacting AA of human SLPI and elafin (PI3) indicated that rat and hamster Slpi2 might not function as elastase inhibitors (Fig 11). In the rabbit, the single *Slpi*

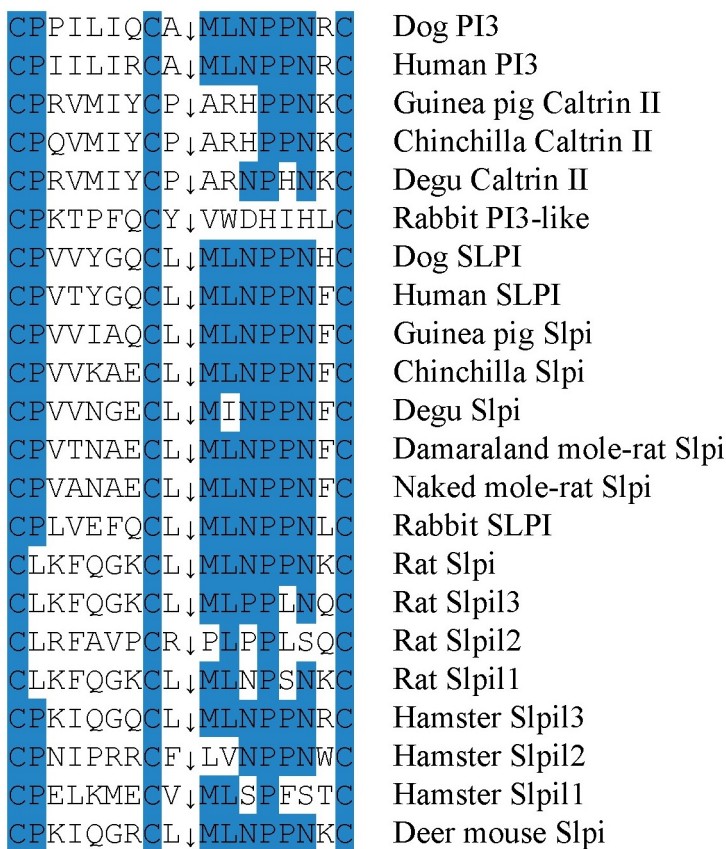

**Fig 11. Conservation of the active site in PI3 and SLPI.** The conserved primary structures at the active site of the human and dog elastase inhibitors PI3 and SLPI are shown together with rabbit and rodent homologs. The scissile bonds are indicated by arrows. Residues that are highlighted in blue are fully conserved at the active site of the human and dog proteins.

was modified and was missing the second coding exon, but the following exon encoded AA indicative of a functional elastase inhibitor (Fig 11).

The rabbit, the pika, the ground squirrel, and the kangaroo rat carried a single *Wfdc15*, whereas in the myomorph and hystricomorph rodents there were two *Wfdc15* –the exception being the guinea pig with one only. The aligned sequences of the protein products displayed a relatively conserved structure, with the characteristic 8 Cys of the WFDC motif. The two genes in the myomorph rodents, *Wfdc15a* and *Wfdc15b*, were structurally similar, but *Wfdc15b* was distinguished in murine and cricetide species by encoding a Cys-Cys duplet in the signal peptide and by lacking the single non-translated nucleotide between the stop codon and the splice donor site downstream of exon 2, which is present in *Wfdc15a* and in non-myomorph *Wfdc15*. The two genes in the hystricomorph rodents, tentatively called *Wfdc15c* and *Wfdc15d*, were best distinguished by different location of conserved Trp residues in the protein products (S36 Fig).

*Wfdc12* was identified by the homology with human and mouse genes. However, there was a major difference between human and mouse *Wfdc12*. The splice donor site 3' to mouse exon 2 is in a different phase than in the human gene and hence a different splice acceptor site is used for exon 3. Analysis of myomorph rodents showed that the rat had the same splice phase at the 3' end of exon 2 as the mouse, whereas the remaining myomorphs had the same splice

phase as human *WFDC12*. However, a differering phase of the splice site was not unique to murine rodents, as it was also found in kangaroo rat *Wfdc12* (S37 Fig).

 *Wfdc5*, which was present with a single copy in all animals of this study, generated the most conserved protein at the locus, *e.g.* 65% of AA residues were identically placed when the primary structure of the rat and guinea pig proteins were compared, as opposed to around 50% when the comparison was made with Slp*i* or the different Wfdc15 variants.

## *Svs1* is confined to myomorph rodents

The *Aoc1* locus, where *Svs1* is situated, was flanked by the conserved genes *Tmem176a* and *Kcnh2* in all of the analyzed rodent species, except for those of the mouse and the rat, where *Kcnh2* was replaced by *Gpnmb*. An extended analysis addressing the location of *Kcnh2* showed that the gene was situated on a different chromosome than *Aoc1* in the mouse and the rat, in contrast to the norm in most mammals, where the two genes are neighbors. This suggested that a recombination between chromosomes had taken place in the region connecting *Aoc1* and *Kcnh2* in the lineage leading to mouse and rat.

 In order to identify *Svs1* and other *Aoc1*-related genes, the genomic sequence spanning the conserved flanking genes was compared to the human *AOC1*. This showed that all analyzed species carried what appeared to be a functional *Aoc1* and that duplication of the gene had taken place only in myomorph rodents (Fig 12). However, the number of duplicated genes varied considerably, being 2 in the jerboa and perhaps as many as 20 in the UGMBMR, most of which appeared to have generated non-functional pseudogenes. *Svs1* was identified in all of the myomorph species, except for the jerboa, which carried 2 *Aoc1* that were 93% similar in sequence and both carrying conserved residues of importance for catalytic activity, suggesting that they generated functional amino oxidases. Of the remaining myomorph species, the hamster, vole, and UGMBMR had frame shift mutations and multiple stop codons, which indicated that *Svs1* was a pseudogene in these animals. The apparently functional genes in mouse, rat, and deer mouse generated differently sized Svs1 molecules due to different size of the repeat region separating the postulated N-terminal domains from the C-terminal domain. There are 7 Lys- and Gln-containing repeats of 18 AA in the mouse, whereas there are 10 in the deer mouse and 12 in the rat. At the C-terminus Svs1 carried the conserved sequence Cys-Val-Cys, which is similar to the sequence Cys-Tyr-Cys found close to the C-terminus in Svs3 and Svsc3.

## TGM4 is mutated in rodents with non-functional SCPs

The conserved genes *Tmem42* and *Zdhhc3* were found to flank the relatively well characterized human and rat *Tgm4* and they were therefore used in order to identify the *Tgm4* locus in species with a less well characterized gene. The genomic DNA sequences spanning *Tmem42* and *Zdhhc3* were probed with the message of human and rat *Tgm4* using the BLAST program with the purpose to determine the structure and potential functionality of the gene. In this way, an apparent functional *Tgm4* consisting of 14 coding exons was identified in 10 of 14 analyzed rodents. The primary structure of rodent Tgm4 was low to moderately conserved, *e.g.* only half of the AA were conserved in the hystricomorph rodents when compared to their myomorph counterparts. Despite this, residues of importance for catalytic activity were strictly conserved (S38 Fig). These residues were also present in the available partial sequence of the kangaroo rat *Tgm4*, suggesting that also the protein of this gene might be functional in spite of the lacking proof from the sequences of exon 10–14, which were missing in the genome database. *Tgm4* of the ground squirrel, naked mole-rat, and Damarland mole-rat were probable pseudogenes. In the ground squirrel the gene was affected by extensive deletions of exons, whereas the genes of

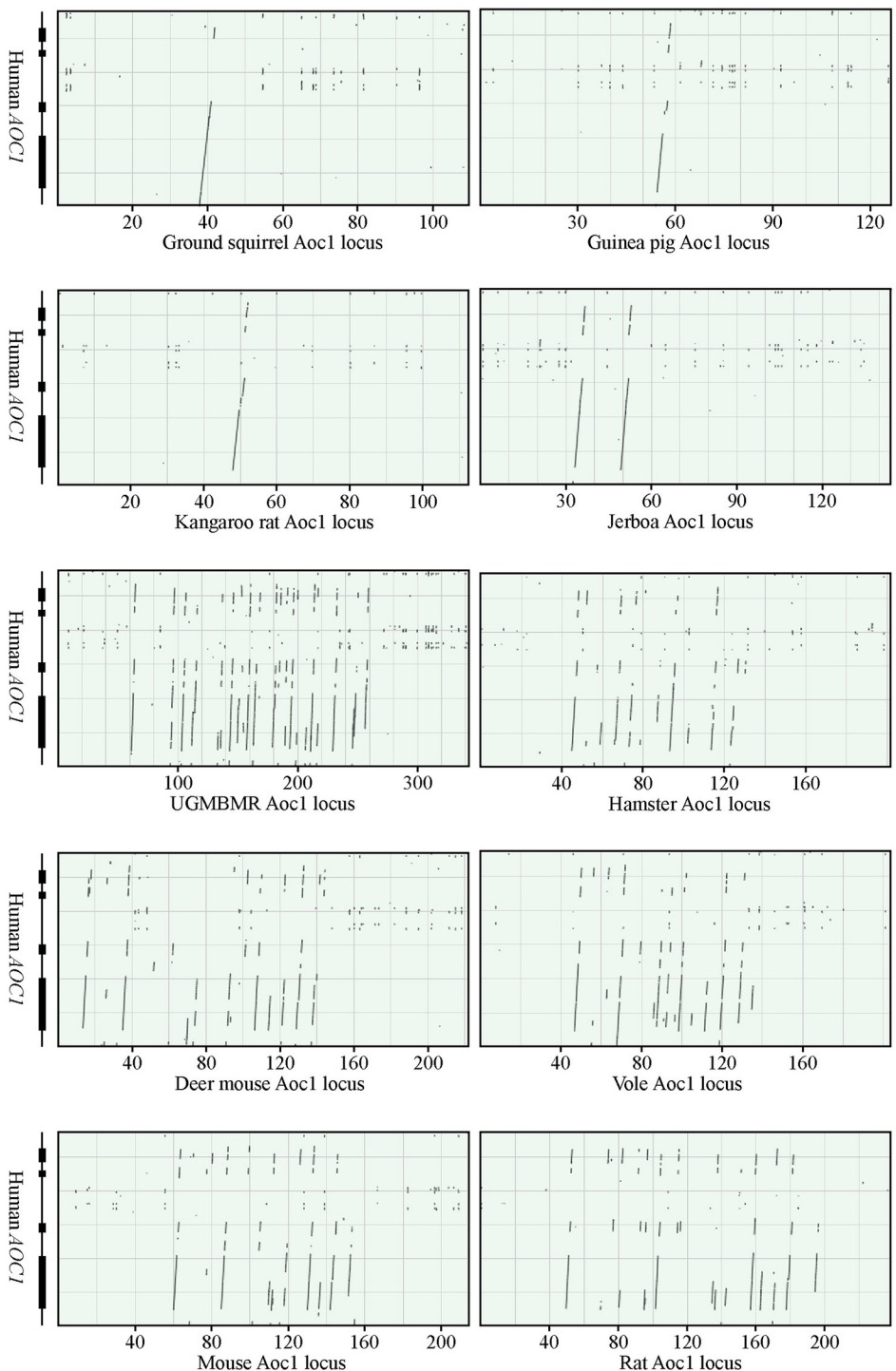

**Fig 12. Duplication of *Aoc1* in myomorph rodents.** The *Aoc1* locus in rodents were analyzed with BLAST in order to detect DNA sequences with homology to the human *AOC1*. The result is displayed in dotplots, with illustrations of the human *AOC1* shown to the right. The Aoc1 is duplicated in all myomorph species, whereas there are no duplication in the non-myomorph species ground squirrel, guinea pig, and kangaroo rat.

the naked mole-rat and Damarland mole-rat carried mutated splice sites or mutations that yielded frameshifts and/or premature stop codons. Similar to the ground squirrel, the two lagomorph species, rabbit and pika, carried pseudogenes with extensive deletion of exons. All species with a non-functional *Tgm4*, except the rabbit, also carried defect genes of the major SCPs.

## Discussion

### Guinea pig Svp5

Early works on guinea pig SVS identified 4 major proteins, which are now known as Svp1-Svp4 [9, 32]. Further studies showed that Svp1 is the major SCP and that it is derived from a precursor molecule by proteolytic processing, which also generates the two almost identical peptides Svp3 and Svp4 [33]. It was therefore a big surprise when we identified Svp5 as a major component in guinea pig SVS. Experiment with TGase showed that it also was a predominating SCP, *i.e.* there are two major SCPs in guinea pig SVS; Svp1 and Svp5. When looking back at older literature, we discovered that a 200 kDa component was identified previously during *in vitro* translation experiments with guinea pig seminal vesicle RNA [9]. We could not find any further information about this component, but most likely it is Svp5. It has perhaps been overlooked in later studies because of poor solubility in some ordinary buffers— we have previously observed that large SCPs have a tendency to form precipitate at storage and handling in commonly used buffers. To overcome the latter, we used a slightly modified version of the high pH/urea-containing buffer, originally devised for the collection of human semen with preserved SEMG1 and SEMG2 [34]. We have previously used it for collecting and diluting the turbid SVS from mouse and rat, in order to avoid losses of SCPs during handling [35]. The buffer turned out to be of outmost importance in the experiments with the rabbit SCP, as rabbit SVS formed a gel-like structure within the glandular tissue, even at normal room temperature. The gel-like structure was easily solved in the high pH/urea buffer and could then be analyzed by SDS-PAGE. The disadvantage with the buffer is that it might affect the conformation of proteins under investigation and if not removed or diluted sufficiently, the urea and high pH might inhibit or affect enzymatic processes used in studies of the SCPs.

During the course of the investigation, we discovered that the structure of Svp5 had been described earlier, but in a different context, under the name guinea pig trappin [36]. The name trappin refers to proteins that similar to PI3 consist of a TSD and a WFDC domain [37]. The guinea pig trappin was discovered during the search for novel trappin molecules and it was also shown that it is a TGase substrate, with high transcript levels in the guinea pig seminal vesicles. Thus, this study confirm the previous results, but with slightly different methodology. In addition, we also show by direct observation, using SDS-PAGE, that the guinea pig seminal vesicles secrete Svp5 at a protein concentration that is comparable to those of Svp1-Svp4, *i.e.* the classical guinea pig SVS proteins. In DNA sequences derived from the guinea pig genome project there is a deletion of 1 bp immediately prior to the WFDC domain in *Svp5*. This leads to a shift of reading frame and an AA sequence that do not encode the WFDC domain. This was not observed in the transcript sequence generated by us or in the previously described guinea pig trappin transcript. There could be several reasons for the discrepancy of course, but 0.1 kb downstream from the position of the deletion there is a gap in the genomic sequence that might also suggest low sequencing coverage of the adjacent sequences. It is therefore not unlikely that the deletion in fact is a sequencing error in the genomic sequence.

The structure of the *Svp5* transcript suggested that it generates a secreted protein consisting of an N-terminal TSD and a C-terminal WFDC domain. This is similar to the structure of PI3, but with important differences: The TSD of Svp5 is 30 times larger than that of human PI3,

but the primary structure flanking the potential scissile bond indicated that the WFDC domain of Svp5 is a poor elastase inhibitor, something that has also been shown elsewhere by experiments on the recombinantly expressed protein [36]. Much of the TSD in Svs5 appears to consist of 24 AA tandem repeats, similar to those in Svp1, but it is very difficult to assign a specific repeat length to large parts of the sequence. The TSD in human PI3 has been described to consist of hexapeptide tandem repeats [20]. Virtually all of the TSD in Svp5 are related to the TSD in PI3, as is show in this paper. Their relationship was also corroborated by the frequent occurrence in Svp5 of the tetrapeptide sequence Val-Lys-Gly-Gln and its derivatives Vla-Lys-Gly and Gly-Gln-Asp, which are conserved peptide sequences at the core of the hexapeptide repeats in the TSD of human PI3.

## Rabbit SVP200

The analysis of rabbit SVS demonstrated that there is a single predominant SCP and no other highly expressed REST gene products in this species. The protein, tentatively denoted SVP200, migrated on SDS-PAGE as if it was larger than guinea pig Svp5, whereas in reality it appears to be smaller. Both of them showed a reduced mobility on SDS-PAGE, but the effect was especially pronounced with SVP200, which displayed an apparent size that was close to 60% larger than anticipated from the AA content. SVP200 and Svp5 are similar in structure, which is dominated by tandem repeats related to the TSD in PI3. Both also contain 2 Cys that presumably form an internal disulfide bond, as suggested by the similar migration on SDS-PAGE in the presence or absence of reducing agents. However, there are also some subtle differences in chemical composition that might be of importance for the electrophoretic behavior, such as a high content in SVP200 of Pro and the acidic residues Glu and Asp, all of which are known to negatively affect the mobility in SDS-PAGE. The 18.3% of acidic residues in secreted SVP200 would be responsible for 22 kDa of the size increase on SDS-PAGE, when calculated according to [38]. The tumor suppressor protein p53 has a high content of Pro, which is considered to yield a molecular mass on SDS-PAGE that is higher than the actual mass calculated from the AA composition [39]. The Pro content of SVP200 is 9,7%—identical to that of mouse p53— and would increase the molecular mass with 29 kDa on SDS-PAGE if the same relative molecular mass increase as for p53 is used. This leaves 23 kDa unaccounted for, but this error is of a relative size that is very similar to the error of 29 kDa that was overestimated by SDS-PAGE for guinea pig Svp5. Another interesting consequence of the high proportion of acidic residues in SVP200 was the relatively low pI of 4.9, which is in sharp contrast to most other SCPs with pI values around 10. This could perhaps indicate that rabbit SVP200 could be substrate for a different TGase than Tgm4, which is considered to be the enzyme for which most SCPs are the substrate.

Analysis of the rabbit genome sequence revealed that the SVP200 gene carried an extra exon, which is not present in other REST genes studied this far. It is located 106 bp upstream to the large exon that codes for most of the secreted protein and generates a peptide sequence of 56 AA that form a unique N-terminal domain in SVP200. Nucleotide sequences homologous with the extra exon are found at a similar position in the first intron of both *Svp5* and *PI3*. Homologous sequences were also found in the first intron of *Svp1* and *Svp2*, but not in any of the other known rodent or primate REST genes. Analysis of the nucleotide sequence collection at NCBI yielded a few potentially homologous sequences with relatively low alignment scores, the majority of which were dominated by *PI3* and related trappin genes. The question whether these homologous sequences were expressed in an ancient *PI3* or is the result of de novo recruitment of an intron sequence to the transcript in rabbits remains to be answered.

The rabbit *SVP200* transcript sequence determined by us is larger than the transcript size expected from the *SVP200* sequence in the rabbit genome database. The size difference is 2 repeats of 72 bp, which probably were formed by duplication of a 144 bp gene segment, as suggested by the presence of a neighboring DNA sequence of that size with only 6 NT difference. Size differences in SCPs due to repeat number polymorphism is relatively common, *e.g.* in primate SEMG1 [11]. Thus the difference in repeat numbers between our sequence and the one derived from the rabbit genome project is not unique among SCPs. The Swedish lop rabbit is not an officially recognized breed, but most likely it is related to the English lop rabbit, which is considered to be the original lop breed that was developed in the early 19th century and which is the founder of many present day lop eared breeds, according to American Rabbit Breeders Association's home page (https://arba.net/recognized-breeds/). The methodology we used to sequence the *SVP200* transcript is expected to identify SNPs, but none was found, suggesting a high degree of homozygosity. However, when we compared our sequence to the one derived from the data base, we found there were unusually many differences between the two sequences, *i.e.* SNPs, particularly in a region located 1.7 to 3.0 kb into our transcript. This region, which holds the 2 extra 72 bp repeats, also carries several perfect matches between sequences that are not located in tandem. We suggest that this might be caused by gene conversion, from which also follows that the accumulation of SNPs in the lop rabbit may not only be caused by random point mutations, but also by a homogenizing process affecting repeats in this region. Interestingly, species specific homogenizing also seemed to affect the two Svs3 genes and the 5' halves of *Svs5* and *Svs6*. This could perhaps reflect a low conservation pressure on the REST gene products. The rapid evolution of the REST genes would then not be caused by strong positive selective forces on the products as such, but rather of low or very specific selective forces. Perhaps the number, spacing, or spatial orientation of critical AA, *e.g.* Gln and Lys, are more important than the overall structure.

## REST genes

Our hypothesis on the origin of the REST genes states that they were created following a duplication of *PI3*, which generated a gene in which the TSD was expanded, in taxon specific processes involving duplication and replication slippage of tandem repeats. It was based on our studies on primate and murine SCPs, where we had shown that SPCE and 3NTE of these genes are related to exon 1 and 3 of *PI3* [19]. The process generated REST genes with highly different structure of the MCE, even in closely related species [16, 18]. The WFDC protease inhibitor moiety was eventually lost, due to lack of selective pressure, to yield the present day primate and murine REST genes. No clear similarities were found between MCE of primate and murine REST genes and exon 2 of *PI3*, except that MCE of the SCPs encoded a TSD, with a relative abundance of Lys and Gln, similar to the TSD in *PI3*. The lack of conserved sequences between *PI3* exon 2 and MCE of primate and murine REST genes was no surprise, given the rapid evolution of the latter, as illustrated by the limited conservation in MCE between human *SEMG1* and mouse *Svp2*, which is confined to the splice donor sites and short sequences close to the termini of the exon [16]. The hypothesis also entails that the REST genes initially evolved in an ancestor of primate and rodent species.

The finding of sequences homologous with coding NT in the MCE of *SEMG2* and exon 1 of *PI3* in the first intron of *Svp5* suggested that the gene was the result of a merger between two genes. This is also in line with what previously has been shown for *Svp1* [22]. Given the similarities between *Svp1*, *Svp2*, *Svp5*, and *Calt2*, the merger between the *SEMG2*-like and a *PI3*-like gene probably took place in a common ancestor within the linage leading to the hystricomorph rodents. Thus, a *PI3*-like gene was recruited a second time in order to generate a SCP.

Presumably this was followed by a duplication that generated one gene evolving to *Calt2* and one to a progenitor of *Svp1*, *Svp2*, and *Svp5*. From this also follows that the gene merger very likely happened after the separation of hystricomorph and myomorph rodents. Thus, a new REST gene was created with the promoter and SPCE from an old REST gene and a *PI3*-like gene with a coding message that evolved to the MCE of a REST gene by duplication of tandem repeats in an analogous fashion as described above for the murine Svs genes and the human semenogelin genes. Rabbit *SVP200* also seem to be the result of a merger between a *SEMG2*-like and a *PI3*-like gene, as nucleotide sequences homologous with exon 2 of *SEMG2* and exon 1 of *PI3* are found in its first intron. There are also sequences homologous with NT encoding the WFDC domain of guinea pig Svp5 and human PI3 located immediately 3' to the SVP200 gene, which presumably were omitted from the gene during evolution due to lack of selection pressure. The discovery of the close similarity between the genes encoding rabbit SVP200 and guinea pig Svp5 was a surprise, as it seems to imply that hystricomorp rodents are more closely related to lagomorph species than they are to myomorph rodents.

To address this further we decided to make an extended investigation of REST gene loci in the mammalian clade Glires. We therefore retrieved and analyzed genome sequences located between the conserved flanking genes *Matn4* and *Kcns1* in rodent and lagomorph species. Sequence analysis demonstrated unique REST genes in myomorph, hystricomorph, and castorimorph rodents, whereas none was found in the analyzed sciuromorph rodent. In the American pika, representing the lagomorph family Ochotonidae, no REST gene was found, similar to the finding in sciuromorph rodents, whereas the rabbit, a lagomorph of the Leporidae family, carried the single functional REST gene SVP200, as discussed above.

The four REST genes we identified in the kangaroo rat, a castorimorph rodent, had features indicating closer relationship to the myomorph REST genes than to those of the hystricomorph suborder. The kangaroo rat REST gene locus also carried Wfdc12 similar to myomorph rodent. This gene was not present in the other suborders and thus further strengthens the close kinship between myomorph and castorimorph rodents.

Analysis of the REST gene loci in rabbit and hystricomorph rodents showed that, in addition to the structural similarities of the SCP genes, the extended rabbit REST gene locus also carried additional composite genes that appears to be the result of merging *SEMG2*- and *PI3*-like genes. In the rabbit there are 4 genes with homology to *SEMG2*, with at least 3 of which are composite genes that show clear homology also to P*I3*. In contrast to the guinea pig, where all of the composite genes are functional, only 2 genes at the rabbit REST gene locus appears to be functional–*SVP200* and the WFDC gene with homology to human PI3 and guinea pig *caltrin2*. In spite of the large size difference—the guinea pig REST gene locus is 2.5 times larger than the rabbit locus—the similarity between them is striking. If the principle of parsimony is to be followed, it is very likely that the original merger between *SEMG2* and *PI3* took place in an ancestor of rabbits and hystricomorph rodents. Most likely, this original gene was duplicated to generate one that developed into a SCP gene and one simple WFDC gene that resembled *PI3*, but with a translation product that was lacking the protease inhibiting specificity of PI3.

The ground squirrel was the only available species of sciuromorph rodents. It was also, together with the pika, the only species lacking functional REST genes at the analyzed locus. It is of course impossible to tell whether this is also true for all members of the suborder sciuromorpha, as well as all pikas, but it gives a hint that this could be the case. However, it does not exclude that there are other SCPs and REST genes in ground squirrels and pikas, or species closely related to them. Such genes could be located outside of the here analyzed locus, in a similar way as Svs1 does in myomorph rodents. The products of such genes could probably be detected by SDS-PAGE on semen or seminal vesicle secretion, as done in this paper and

previously on mouse, rat, and human specimens. In those analyses, all REST gene products could be linked to the REST gene locus analyzed in this paper, except for mouse and rat *Svs1*. This gene has not evolved from a gene of a known TGase substrate. Instead it evolved from *Aoc1*, a gene that encodes a $Cu^{2+}$ containing amino oxidase with diamines and polyamines as preferred substrates. However, similar to the SCPs encoded at the REST gene locus, Svs1 has developed a repeat region, with varying copy numbers in different species, but in contrast to Svs2, the size of these repeats did not seem to vary and the repeat structure was also fairly conserved. *Svs1* evolved following duplication of *Aoc1*, something that seems to have occurred only in myomorph rodents. In this rodent suborder *Aoc1* has been duplicated several fold, generating many truncated pseudogenes, but also *Svs1* and at least 2 additional genes that appear to be functional. We described them in our paper on the Svs1 structure and tentatively named them diamine oxidase like 1 and 2, abbreviated Doxl1 and Doxl2 [17]. They are presumably lacking amino oxidase activity, as they similar to Svs1 are missing either the critical Tyr that generates the catalytic topaquinone or one or more $Cu^{2+}$ coordinating His. Because of a very low conservation of the protein paralogs in mouse and rat, we speculated in our paper that the genes were relatively old. This was wrong: the finding in this study, showing that duplications of *Aoc1* are confined to myomorph rodents, instead shows that *Svs1* and its homologs are evolutionary young. The low conservation of the protein products is instead caused by a very rapid evolution.

## SCPs and TSD

Early works on semen coagulation in rodents demonstrated that a copulatory plug developed in the vagina following mating because a prostate-secreted TGase acted to crosslink seminal vesicle-secreted coagulum proteins, as discussed in [2]. The copulatory plug is considered to act as a barrier to sperm from a second mating male. However, the prostate secreted TGase, now known as TGM4, is expressed in the prostate of many species, some of which are known not to form a copulatory plug, *e.g.* humans. It is well known that semen is a rich source of polyamines which potentially can modify Gln in peptides and proteins by TGase- mediated isopeptide formation. It has been suggested that such modifications could be of importance to suppress immune response to male components in the female genital tract by modifying immune determinants [40]. There could also be other functions of TGM4 that are presently not known.

In this paper we observe a direct link between the major SCPs and TGM4 in rodents, as those species, which are lacking a functional SCP, are also lacking a functional TGM4. Two of these species, the naked mole-rat and Damarland mole-rat, are eusocial mammals living in colonies with a single mating female and one or a few mating males. It is easy to imagine that the selection pressure for copulatory plug formation would be low under such circumstances, and hence also for SCPs and TGM4. The same explanation could not be used for the ground squirrel, the third rodent lacking both SCPs and TGM4. What distinguish it from rodents like rat, mouse, and guinea pig is that it is a solitary animal with a single annual breeding season, but whether such behavioral parameters has anything to do with the lack of SCPs and TGM4 in this species is far from clear. Another interesting finding was that the rabbit is lacking TGM4, in spite of its ability to secrete high amounts of a SCP. Perhaps, the rabbit has replaced TGM4 by overexpressing a different TGase in the prostate. This could then also be the reason behand the unique properties of SVP200 compared to other SCPs, e.g. the low pI and the N-terminal domain.

## Conclusions

In this paper we describe two major semen coagulum proteins, with structure and similarities suggesting a closer relationship between hystricomorph rodents and lagomorph species than to myomorph and castorimorph rodents. We also show that Svs1, a major SCP in mouse and rat, is confined to myomorph rodents. Finally, we also show that rodents lacking a functional SCP frequently also have silenced the prostate expressed *TGM4*. Further studies will be directed against SCPs and other REST genes in the full complement of mammal orders to achieve a more comprehensive picture on the origin and evolution of this interesting gene family.

## Supporting information

**S1 Table. Primer used in sequencing of the guinea pig *Svp5* transcript.**
(DOCX)

**S2 Table. Primer used in sequencing of the lop rabbit SVP200 transcript.**
(DOCX)

**S1 Fig. Conformity of guinea pig *Svp5* transcript and peptide sequences.** The cDNA sequence of *Svp5* is shown with the translated protein sequence written above. The predicted signal peptide is highlighted in grey. Peptide sequences highlighted in yellow indicates that they agree with peptides generated by trypsin digestion of the 190 kDa component in the guinea pig seminal vesicle secretion.
(DOCX)

**S2 Fig. Polymorphisms in the guinea pig *Svp5*.** The nucleotide sequence of the *Svp5* transcript was compared to the guinea pig genome sequence and the trappin transcript by BLAST. Nucleotide differences are highlighted in grey and amino acid substitutions are highlighted in purple. The numbers to the right refers to the position in the *Svp5* transcript.
(DOCX)

**S3 Fig. Conformity of rabbit SVP200 transcript and peptide sequences.** The cDNA sequence of SVP200 is shown with the translated protein sequence written above. The predicted signal peptide is highlighted in grey. Peptide sequences highlighted in yellow indicates that they agree in molecular mass with peptides generated by trypsin digestion of the 200 kDa component in the rabbit seminal vesicle secretion. A peptide occurring 3 times in SVP200 is underlined.
(DOCX)

**S4 Fig. Differences between the SVP200 transcript and the sequence in the rabbit genome database.** The SVP200 transcript from the lop eared rabbit is given with non-translated nucleotides highlighted in grey. A block of 144 nucleotides, not present in the genome database, is highlighted in blue and an adjoining, almost identical, sequence is highlighted in yellow. A CT dinucleotide in the microsatellite, located in the 3' non-translated sequence, which is not present in the genome sequence, is also highlighted in blue. Nucleotide positions in the SVP200 transcript that differ from those in the rabbit genome database are highlighted purple.
(DOCX)

**S5 Fig. Homology of human *PI3* and DNA sequences flanking the 3' end of rabbit *SVP200*.** The aligned sequences are shown with vertical bars indicating conserved nucleotides. The sequences highlighted in grey are 3' non-translated nucleotides and those highlighted in green are translated nucleotides in exon 2 of PI3, with the translation written below. The stop codon

is highlighted in red and the polyadenylation signals are underlined.
(DOCX)

**S6 Fig. Dotplots showing conserved DNA sequences in human PI3 and rat REST genes.**
Sequences were aligned using the computer program BLAST, which also generated dotplots
that illustrated the location of conserved nucleotides. The compared sequences consisted of
the genes and 200 bp of 5' and 3' gene flanking DNA. Human *PI3* is illustrated as a horizontal
line with boxes showing, from left to right, the location of exons 1–3.
(PDF)

**S7 Fig. Alignment of conserved DNA sequences in human PI3 and rat REST genes.** The
aligned sequences are shown with conserved nucleotides indicated by vertical bars. Exon
sequences are highlighted in green if they are translated or in grey if they are non-translated.
(DOCX)

**S8 Fig. Dotplots showing homology of human SEMG2 and REST genes in myomorph
rodents.** The dotplos were generated by the sequence alignment program BLAST. Compared
sequences consisted of genes and 200 bp flanking DNA at both ends. Below the dotplots is a
schematic illustration of SEMG2, with boxes showing, from left to right, the approximate loca-
tion of exons 1–3, *i.e.* SPCE, MCE, and 3NTE.
(PDF)

**S9 Fig. Dotplots showing homology of human SEMG2 and REST genes in myomorph
rodents.** The dotplos were generated by the sequence alignment program BLAST. Compared
sequences consisted of genes and 200 bp flanking DNA at both ends. Below the dotplots is a
schematic illustration of SEMG2, with boxes showing, from left to right, the approximate loca-
tion of exons 1–3, *i.e.* SPCE, MCE, and 3NTE.
(PDF)

**S10 Fig. Dotplots showing homology of human SEMG2 and REST genes in myomorph
rodents.** The dotplos were generated by the sequence alignment program BLAST. Compared
sequences consisted of genes and 200 bp flanking DNA at both ends. Below the dotplots is a
schematic illustration of SEMG2, with boxes showing, from left to right, the approximate loca-
tion of exons 1–3, *i.e.* SPCE, MCE, and 3NTE.
(PDF)

**S11 Fig. Dotplots showing homology of human SEMG2 and REST genes in myomorph
rodents.** The dotplos were generated by the sequence alignment program BLAST. Compared
sequences consisted of genes and 200 bp flanking DNA at both ends. Below the dotplots is a
schematic illustration of SEMG2, with boxes showing, from left to right, the approximate loca-
tion of exons 1–3, *i.e.* SPCE, MCE, and 3NTE.
(PDF)

**S12 Fig. Dotplots showing homology of human SEMG2 and REST genes in myomorph
rodents.** The dotplos were generated by the sequence alignment program BLAST. Compared
sequences consisted of genes and 200 bp flanking DNA at both ends. Below the dotplots is a
schematic illustration of SEMG2, with boxes showing, from left to right, the approximate loca-
tion of exons 1–3, *i.e.* SPCE, MCE, and 3NTE.
(PDF)

**S13 Fig. Conservation in myomorph rodents of DNA sequences surrounding the MCE
splice sites in human *SEMG2*.** The DNA sequences were aligned using the computer program
Clustal Omega, which was followed by minor manual adjustments of the aligned sequences

Translated nucleotides are highlighted in green and non-translated in grey.
(DOCX)

**S14 Fig. Conservation in myomorph rodents of DNA sequences surrounding the MCE splice sites in human *SEMG2*.** The DNA sequences were aligned using the computer program Clustal Omega, which was followed by minor manual adjustments of the aligned sequences Translated nucleotides are highlighted in green and non-translated in grey.
(DOCX)

**S15 Fig. Conservation in myomorph rodents of DNA sequences surrounding the MCE splice sites in human *SEMG2*.** The DNA sequences were aligned using the computer program Clustal Omega, which was followed by minor manual adjustments of the aligned sequences Translated nucleotides are highlighted in green and non-translated in grey.
(DOCX)

**S16 Fig. Conservation in myomorph rodents of DNA sequences surrounding the MCE splice sites in human *SEMG2*.** The DNA sequences were aligned using the computer program Clustal Omega, which was followed by minor manual adjustments of the aligned sequences Translated nucleotides are highlighted in green and non-translated in grey.
(DOCX)

**S17 Fig. Conservation in myomorph rodents of DNA sequences surrounding the MCE splice sites in human *SEMG2*.** The DNA sequences were aligned using the computer program Clustal Omega, which was followed by minor manual adjustments of the aligned sequences Translated nucleotides are highlighted in green and non-translated in grey.
(DOCX)

**S18 Fig. Conservation of REST genes encoding small protein products.** Dotplots generated by BLAST alignment of *Svs4*, *Svs5*, and *Svs6* from deer mouse and rat. The illustrations below and beside the axis depict the genes, with rectangles indicating the location of exons.
(PDF)

**S19 Fig. Primary structure alignment of Svs2.** The amino terminal tandem repeats and the extended central repeat region were removed from the sequences, which were then aligned with Clustal Omega, followed by some minor manual adjustment. The upper part shows the aligned sequences with amino-terminal tandem repeats, highlighted in green or blue, reinserted. The underlined sequences are encoded by SPCE (exon1) and the conserved Cys preceding the central repeat regions are highlighted in grey. A frame shifted sequence, leading to premature stop, in the deer mouse is written with red font. The lower part displays the central repeat regions grouped according to suborder. The sequences were manually aligned in order to demonstrate conserved sequences both within and at the termini of the repeats. Some of the longer, perfectly conserved repeats are underlined. The hydrophobic AA, surrounded by Gln and Lys at the center of the poorly defined tandem repeats are highlighted in red and similar repeats with a different central residue are highlighted in purple. Aromatic residues are highlighted in yellow.
(DOCX)

**S20 Fig. Primary structure alignment of murine and cricetide Svs3.** Sequences were aligned with Clustal Omega. The sequences encoded by SPCE are underlined and conserved Cys are highlighted in grey. Tripeptides sequences of Gln and Lys surrounding a hydrophobic or a non-hydrophobic residue are highlighted in red and purple respectively.
(DOCX)

**S21 Fig. The extended central repeat region of Svs3b in UGMBMR.** The primary structures were manually aligned after removal of the central repeat regions. Stars (*) and colon (:) indicates fully and 3 out of 4 conserved residues respectively. Gln and Lys surrounding a hydrophobic or a non-hydrophobic residue are highlighted in red and purple respectively.
(DOCX)

**S22 Fig. Unique intron in murine Svs6.** Aligned sequences are shown with exon 2 and the beginning of exon 3 in rat and mouse highlighted in green. Positions with homology to the murine splice donor or acceptor site, or both, are mutated in non-murine genes. Potentially alternative splice donor and acceptor sites are highlighted in grey and purple respectively. The underlined sequence is the last of 6 tandem repeats in vole Svs6; for clarity were the first 5 omitted. Identical nucleotide in 5 or 6 species is indicated with a star (*).
(DOCX)

**S23 Fig. Alignments of Svs4, Svs5, and Svs6.** The aligned products of functional genes are shown. Residues are highlighted in green if they are present in >50% of the sequences, *i.e.* 9 or more. The sequences highlighted in grey, in deer mouse Svs5 and Svs6, are in a different reading frame, due to conserved mutations. Sequences encoded by SPCE are depicted by a thick underlining. The sequence with the double underlining is the first copy of 6 tandem repeats present in vole Svs6. The remaining repeats were omitted prior to alignment. The complete tandem repeat region of vole Svs6 is shown below the aligned sequences.
(DOCX)

**S24 Fig. Homology of *Svp1*, *Svp2*, and *Svp5* with human *SEMG2* and *PI3*.** Dotplots were generated by the alignment program BLAST. The genes are outlined below and beside the dotplots, with location of exons illustrated by boxes.
(PDF)

**S25 Fig. Homology of *Svp1*, *Svp2*, and *Svp5* with human *SEMG2* and *PI3*.** Dotplots were generated by the alignment program BLAST. The genes are outlined below and beside the dotplots, with location of exons illustrated by boxes.
(PDF)

**S26 Fig. Homology of *Svp1*, *Svp2*, and *Svp5* with human *SEMG2* and *PI3*.** Dotplots were generated by the alignment program BLAST. The genes are outlined below and beside the dotplots, with location of exons illustrated by boxes.
(PDF)

**S27 Fig. Retainment of a second intron in *Svp5*.** The 3' end of human *PI3* was aligned with homologous parts in *Svp5*. Translated nucleotides are highlighted in green and the stop codons in red. The known 3' non-translated NT in *PI3* and guinea pig *Svp5* are highlighted in grey. As can be seen, the spice donor site is mutated in the guinea pig and the degu, and the spice acceptor site in the degu.
(DOCX)

**S28 Fig. Sequence alignment of hystricomorph rodents Svp2.** The aligned primary structure of Svp2 precursors are shown with star symbols (*) indicating residues conserved in all of the analyzed species. Underlined residues are encoded by SPCE.
(DOCX)

**S29 Fig. Sequence alignment of hystricomorph rodent Svp1 precursors.** Sequence alignment of translations from hystricomorph *Svp1*, including potential pseudogenes, are shown.

Damaraland and naked depict the mole-rats. Star symbols (*) highlighted in red indicate stop codons, whereas the same symbol written below the aligned sequences indicates fully conserved residues. The underlining at the amino-terminals depict residues encoded by SPCE and the underlining central in the guinea pig sequence shows the location of the processing site that generates Svp3/4 from the N-terminal part of the precursor and Svp1 from the C-terminal part. The residues highlighted in yellow show conserved residues, which suggest that the tandem repeats in Svp3/4 and Svp1 are homologous and probably have evolved following an initial duplication of a peptide encompassing Trp, Lys, and Gln.
(DOCX)

**S30 Fig. Sequence alignment of hystricomorph rodent Svp5.** Alignments of the highly conserved termini in hystricomorp Svp5 is shown. Conserved residues are indicated by star symbols (*).
(DOCX)

**S31 Fig. Kangaroo rat REST genes.** Nucleotide sequences of the genes, tentatively denoted Svsc1-Svsc4, are given with translated nucleotides highlighted in green, and non-translated in grey. The TATA box in the upstream promoter region is doubly underlined and translations in one-letter code are written above the coding nucleotides. Two poly-Gln tracts in Svsc3 are highlighted with thick underlining.
(DOCX)

**S32 Fig. Kangaroo rat REST genes.** Nucleotide sequences of the genes, tentatively denoted Svsc1-Svsc4, are given with translated nucleotides highlighted in green, and non-translated in grey. The TATA box in the upstream promoter region is doubly underlined and translations in one-letter code are written above the coding nucleotides. Two poly-Gln tracts in Svsc3 are highlighted with thick underlining.
(DOCX)

**S33 Fig. Kangaroo rat REST genes.** Nucleotide sequences of the genes, tentatively denoted Svsc1-Svsc4, are given with translated nucleotides highlighted in green, and non-translated in grey. The TATA box in the upstream promoter region is doubly underlined and translatations in one-letter code are written above the coding nucleotides. Two poly-Gln tracts in Svsc3 are highlighted with thick underlining.
(DOCX)

**S34 Fig. Kangaroo rat REST genes.** Nucleotide sequences of the genes, tentatively denoted Svsc1-Svsc4, are given with translated nucleotides highlighted in green, and non-translated in grey. The TATA box in the upstream promoter region is doubly underlined and translatations in one-letter code are written above the coding nucleotides. Two poly-Gln tracts in Svsc3 are highlighted with thick underlining.
(DOCX)

**S35 Fig. Alignment of kangaroo rat REST gene products.** The primary structure of Svsc1 aligned with that of Svsc2 are shown at the top. Below, Svsc4 is shown aligned with Svsc3, from which a central repeat region has been removed. Conserved residues are indicated by star symbols (*). Highlighted in grey are the conserved Cys, which are homologous with equally placed residues in myomorph Svs2 and Svs3. The Lys-Gly dipeptides in the C-terminal half of Svsc4 are highlighted in yellow and the tandem repeats are underlined. The central repeat region in Svsc3 is shown at the bottom, with the Gln-Leu-Lys motif highlighted in red.
(DOCX)

**S36 Fig. Alignment of rodent and lagomorph Wfdc15.** The alignment was done with Clustal Omega using default settings, followed by a minor manual adjustment. The residues shown N-terminal to the sign (<>) are encoded by exon 1 and those C-terminal to the sign are encoded by exon 2. The Cys duplets highlighted in red are in this setting specific for Wfdc15b in myomorph rodents. The location of the Trp highlighted in green separates Wfdc15c from Wfdc15d in hystricomorph rodents. NMR and DMR are abbreviations for naked mole-rat and Damaraland mole-rat.
(DOCX)

**S37 Fig. Differing splice phase in rodent *Wfdc12*.** Sequences surrounding the splice donor site in *WFDC12* are aligned with exon sequences highlighted in green and translations written in one-letter code. As can be seen, the splice phase is the same in mouse and kangaroo rat, which differ from the splice phase in jerboa and human *WFDC12*. The other analyzed rodent and lagomorph *WFDC12* had the same phase as the human gene, except for the rat, which had the same phase as the mouse and kangaroo rat genes.
(DOCX)

**S38 Fig. Conservation of rodent Tgm4.** The aligned primary structures of apparently functional rodent Tgm4 are shown. The amino terminals were heterogeneous due differences in the location of start codon. Residues at the catalytic site are highlighted in blue and additional residues of importance for the catalytic activity of transglutaminases are highlighted in yellow. The fullength sequence of kangaroo rat was not available, but residues of importance for catalysis are present, as can be seen.
(DOCX)

**S1 Raw images.**
(PDF)

# Acknowledgments

We are in deep gratitude to Dr Nishtman Dizeyi, Lund University and Dr. Benjamin Pippenger, Straumann, Basel, Switzerland for providing seminal vesicle secretion and tissue samples for this study.

# Author Contributions

**Conceptualization:** Åke Lundwall, Magnus Jonsson.

**Data curation:** Åke Lundwall.

**Formal analysis:** Karin Hansson.

**Funding acquisition:** Åke Lundwall.

**Investigation:** Åke Lundwall, Margareta Persson, Karin Hansson.

**Supervision:** Åke Lundwall.

**Visualization:** Åke Lundwall.

**Writing – original draft:** Åke Lundwall.

**Writing – review & editing:** Magnus Jonsson.

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
