## [Decision Letter · Decision Letter 0]

18 Aug 2020

PONE-D-20-19168

Identification of the major rabbit and guinea pig semen coagulum proteins and description of the diversity of the REST gene locus in the mammalian grandorder glires

PLOS ONE

Dear Dr. Lundwall,

Thank you for submitting your manuscript to PLOS ONE. After careful consideration, we feel that it has merit but does not fully meet PLOS ONE’s publication criteria as it currently stands. Therefore, we invite you to submit a revised version of the manuscript that addresses the points raised during the review process.

The manuscript idea and the data are good. However, it should be organized and shortened if possible with clear sub-headings. Also authors should provide sufficient information including software packages/Databases etc. for the work to be reproduced independently.

We look forward to receiving your revised manuscript.

Kind regards,

Academic Editor

PLOS ONE

Additional Editor Comments:

The manuscript idea and the data are good. However, it should be organized and shortened if possible with clear sub-headings. Also authors should provide sufficient information including software packages/Databases etc. for the work to be reproduced independently.

Journal Requirements:

Reviewers' comments:

Reviewer's Responses to Questions

**Comments to the Author**

1. Is the manuscript technically sound, and do the data support the conclusions?

Reviewer #1: Partly

Reviewer #2: Yes

Reviewer #3: Yes

2. Has the statistical analysis been performed appropriately and rigorously? 

Reviewer #1: N/A

Reviewer #2: Yes

Reviewer #3: I Don't Know

3. Have the authors made all data underlying the findings in their manuscript fully available?

Reviewer #1: Yes

Reviewer #2: Yes

Reviewer #3: Yes

4. Is the manuscript presented in an intelligible fashion and written in standard English?

Reviewer #1: No

Reviewer #2: Yes

Reviewer #3: Yes

5. Review Comments to the Author

Reviewer #1: In the present article titled “Identification of the major rabbit and guinea pig semen coagulum proteins and description of the diversity of the REST gene locus in the mammalian grandorder glires” the author identified uncharacterized semen coagulum proteins Svp5 and Svp200 in the seminal vesicle secretions of guinea pig and rabbit respectively. Author also identified the corresponding genes for these two proteins by homology comparison with human and murine genomes. According to the author, both Svp5 and Svp200 are formed as a result of two gene merging which are homologous with human SEMG2 and PI3 genes. In addition to this, the author also explained the diversity of REST gene locus in the mammalian clade “Glires”.

Following are my comments.

1) This manuscript has two parts (a) Identification and characterizing Svp5 and Svp200 in the seminal vesicle secretions of guinea pig and rabbit respectively. (b) The diversity of REST gene locus in the mammalian clade “Glires”. I am not much convinced by the second part of the manuscript. Based on sequence homologies of REST gene locus author concluded that hystricomorpha separated from myomorpha and castorimorpha before the separation of hystricomorpha from lagomorpha.

2) The manuscript is too lengthy and it is too complicated at several places particularly in the second part of the story. I suggest the author to condense the manuscript and make it smile.

3) The Material and method section do not have subheadings and it’s not organized properly. It is always good to have separate sub heading in the methods section.

4) According to me the diversity of REST gene locus in the mammalian clade “Glires”, the author has done deeper study and complicated the study. I think that part of the manuscript can go separately.

Reviewer #2: This manuscript is focused on to identify two major SCPs, one in rabbit and one in guinea pig, and sequenced their cDNA. To understand the origin and evolution of the SCPs author also investigates the REST gene locus in the mammalian grandorder glires by analysing DNA sequences retrieved from genome databases.

The key points of this study are structural characterization rabbit SVP200 and guinea pig Svp5 gene and the close similarity between them seems to imply that hystricomorp rodents are more closely related to lagomorph species.

Overall the quality of the data is very good, and the writing is clear and organized. I am fully convinced with results and discussion. I also expect that the information in this manuscript should be useful for researcher involved in such studies.

Reviewer #3: The manuscript is about interesting identification of rabbit and guinea pig semen coagulum proteins and the cDNA being sequenced and further comprehensive studies on REST gene evolution. The manuscript is extensively written however the structure of the manuscript can be modified and shortened to a certain degree for the ease of reading.

1. A flow chart of how sequentially the analysis is done from initial conception to conclusion can be made and included in the initial figure.

2. Kindly organize the methods with proper subtitles of the corresponding experiments. Mention the software used for making the dotplot alignment from BLAST output ?

3. More emphasis can be given to the major biological impact of the current study can be described little bit in introduction or discussion.

4. What is the significance of identification of SvP5 in guinea pigs over other SvP. Did authors try to characterize the functional role of SvP5 ?

5. Explain the link between SCPs and TGM4 in rodents how they can have functional impacts in rodents , what avenues the new study will open up for further analysis.

6. A model figure or a molecular phylogenetic analysis of REST family would be interesting to see overall connection with each other species.

6. PLOS authors have the option to publish the peer review history of their article (what does this mean?). If published, this will include your full peer review and any attached files.

Reviewer #1: No

Reviewer #2: No

Reviewer #3: No

---

## [Author Response · Author response to Decision Letter 0]

17 Sep 2020

Response to the Academic Editor and the Reviewers

Academic Editor

We have considered the reviewers’ suggestions and made changes to the manuscript. The title has also been slightly modified to ‘Identification of the major rabbit and guinea pig semen coagulum proteins and description of the diversity of the REST gene locus in the mammalian clade Glires’. We did not feel that the manuscript could be shortened without losing vital information, but we have introduced new subheadings that we think will improve the readability of it. There is sufficient information in the manuscript to independently reproduce our experiments. The information regarding software and databases was in our original submission unfortunately placed in a context, such that it could lead to confusion and misunderstandings. It has now been moved to a more appropriate place.

Reviewer 1

The reviewer refers to Glires as clade Glires, even though it has also been ranked as a grandorder. After careful consideration, we have decided to use the term clade in order to describe Glires. Hence, the title of the paper has also been changed.

Response to comment 1

Our data undoubtedly show that a new gene was created by the merger of SEMG2-like and PI3-like genes in ancestors of both hystricomorph species and rabbits. This could, of course, have happened by two independent events, but according to the principle of parsimony, it is more likely that it happened in a common ancestor. As there were no signs of a similar gene merger in any of the other analyzed species, it also seems more likely that the merger happened after the separation of hystricomorpha from myomorpha and castorimorpha. We are fully aware that this might be controversial and in an effort to avoid being provocative, we have changed the wording of the last sentence in the abstract from ‘hystricomorpha separated from myomorpha’ to ‘hystricomorpha may have separated from myomorpha’.

Our study also hints at a close relation between sciuromorph species and pika, both of which are lacking functional SCPs (and also Wfdc12). This is not discussed in our paper as we deemed it too speculative. However, both this finding and the gene merger in hystricomorpha and rabbit suggest that it might be worthwhile to critically scrutinize the phylogeny of Glires. This is not our area of expertise, but it is our hope that the findings in this paper will stimulate, not only to studies on SCPs evolution, but also to extended studies on the phylogeny of Glires.

Response to comment 2 and 3

The manuscript is relatively, but not extremely, long. We do not feel that it is verbosely written. Instead, we think that the length very much reflects the content and most likely condensation would mean removal of interesting parts of the presentation. We do not agree to this – see also our response to comment 4. To ease up the reading, we have introduced subheadings in the Material and Methods and Discussion sections and also introduced new or modified old subheadings in the Results section.

Response to comment 4

In our view, reporting only the structures of Svp5 and SVP200 would generate a thin paper. Questions regarding the relationship between Svp5 and SVP200 also prompted us to widen the study and analyze REST genes in all rodents and lagomorphs that we could find in sequence databases. This study showed that 3 out of 4 rodent suborders had unique sets of REST genes, whereas the forth suborder seemed to be lacking them. We also noted that the SCPs were pseudogenes in eusocial rodents, which led us to also investigate the prostate transglutaminase TGM4, which in turn showed that all species lacking a functional SCP also carried a non-functional TGM4. We think that by extending the investigation, we have made discoveries that have transformed an ordinary paper to a very interesting paper that we have no intension to split up.

Reviewer 3

Response to comment 1

There is a flow chart on lines 316-324 in the manuscript. We think that moving that flow chart into Figure 1 would only complicate the figure and perhaps also the reading.

Response to comment 2

Subheadings have been introduced in Material and Methods. 

BLAST is perhaps best known as a tool to search for homologies in sequence databases, but it can also be used to compare two sequences. If the web-based program at NCBI is used for the latter purpose, the output is aligned sequences and a dotplot. Dotplots created that way were saved and used in our paper. We think the procedure is appropriately described in the Material and Methods section, but we can also understand that it could lead to misunderstandings and confusion because of where it the text it was placed. It has been moved to a new location under the subheading ‘Study of the REST gene locus in rodents and lagomorphs’.

Response to comments 3-5

The biological function is, of course, an important aspect. Experiments showing that both Svp5 and SVP200 are excellent substrates of transglutaminase were made, but apart from that, function was not primarily addressed in this study. At the end of discussion (line 1068-1081) we address some functional issues, e.g. barrier function of SCPs.

Response to comment 6

We are presently not ready to sketch a hypothetical phylogenetic three, as there are too many options. Classic molecular phylogenetics is, as far as we understand, out of the question when it comes to comparing rodent suborder and lagomorpha, because of the vast sequence differences. It is not an easy task even within suborders, partly because of large sequence differences, but also because of non-random mutations, e.g. the homogenizing process affecting myomorph Svs3, Svs5, and Svs6.

---

## [Decision Letter · Decision Letter 1]

30 Sep 2020

Identification of the major rabbit and guinea pig semen coagulum proteins and description of the diversity of the REST gene locus in the mammalian clade Glires

PONE-D-20-19168R1

Dear Dr. Lundwall,

We’re pleased to inform you that your manuscript has been judged scientifically suitable for publication and will be formally accepted for publication once it meets all outstanding technical requirements.

Kind regards,

Rajakumar Anbazhagan, Ph. D.

Academic Editor

PLOS ONE

Additional Editor Comments (optional):

The authors addressed most of the comments well in detail and its acceptable for publication.

Reviewers' comments:

Reviewer's Responses to Questions

**Comments to the Author**

1. If the authors have adequately addressed your comments raised in a previous round of review and you feel that this manuscript is now acceptable for publication, you may indicate that here to bypass the “Comments to the Author” section, enter your conflict of interest statement in the “Confidential to Editor” section, and submit your "Accept" recommendation.

Reviewer #1: All comments have been addressed

Reviewer #2: All comments have been addressed

Reviewer #3: All comments have been addressed

2. Is the manuscript technically sound, and do the data support the conclusions?

Reviewer #1: Yes

Reviewer #2: Partly

Reviewer #3: Partly

3. Has the statistical analysis been performed appropriately and rigorously? 

Reviewer #1: Yes

Reviewer #2: Yes

Reviewer #3: I Don't Know

4. Have the authors made all data underlying the findings in their manuscript fully available?

Reviewer #1: Yes

Reviewer #2: Yes

Reviewer #3: Yes

5. Is the manuscript presented in an intelligible fashion and written in standard English?

Reviewer #1: Yes

Reviewer #2: Yes

Reviewer #3: Yes

6. Review Comments to the Author

Reviewer #1: In this revised version of the manuscript, The author addressed all my concerns.

My recommendation is to accept with out any further changes.

Reviewer #2: I am satisfied with revised manuscript. Author successfully addressed majority of reviewers comments.

Reviewer #3: (No Response)

7. PLOS authors have the option to publish the peer review history of their article (what does this mean?). If published, this will include your full peer review and any attached files.

Reviewer #1: No

Reviewer #2: No

Reviewer #3: No

---

## [Editor Report · Acceptance letter]

2 Oct 2020

PONE-D-20-19168R1 

Identification of the major rabbit and guinea pig semen coagulum proteins and description of the diversity of the REST gene locus in the mammalian clade Glires 

Dear Dr. Lundwall:

I'm pleased to inform you that your manuscript has been deemed suitable for publication in PLOS ONE. Congratulations! Your manuscript is now with our production department. 

Kind regards, 

on behalf of

Dr. Rajakumar Anbazhagan 

Academic Editor

PLOS ONE